

# Improved pseudolikelihood regularization and decimation methods on non-linearly interacting systems with continuous variables

**Alessia Marruzzo** [1,2⋆], **Payal Tyagi** [2], **Fabrizio Antenucci** [2,3], **Andrea Pagnani** [4,5] **and Luca Leuzzi** [2,6]

**1** Supercomputing Innovation and Application Department, CINECA, Bologna, Italy
**2** Soft and Living Matter Lab., Rome Unit of CNR-NANOTEC, Institute of Nanotechnology, Piazzale Aldo Moro 5, I-00185, Rome, Italy
**3** Institut de Physique Théorique, CEA, Université Paris-Saclay, F-91191, Gif-sur-Yvette, France.
**4** IIGM - Italian Institute for Genomic Medicine, via Nizza 52, 10126, Torino, Italy
**5** Human Genetic Foundation (HuGeF-Torino), Via Nizza 52 I-10126 Torino - Italy
**6** Dipartimento di Fisica, Università *Sapienza*, Piazzale Aldo Moro 5, I-00185, Rome, Italy

⋆ a.marruzzo@cineca.it

## Abstract

We propose and test improvements to state-of-the-art techniques of Bayeasian statistical inference based on pseudolikelihood maximization with $\ell_1$ regularization and with decimation. In particular, we present a method to determine the best value of the regularizer parameter starting from a hypothesis testing technique. Concerning the decimation, we also analyze the worst case scenario in which there is no sharp peak in the tilded-pseudolikelihood function, firstly defined as a criterion to stop the decimation. Techniques are applied to noisy systems with non-linear dynamics, mapped onto multi-variable interacting Hamiltonian effective models for waves and phasors. Results are analyzed varying the number of available samples and the externally tunable temperature-like parameter mimicing real data noise. Eventually the behavior of inference procedures described are tested against a wrong hypothesis: non-linearly generated data are analyzed with a pairwise interacting hypothesis. Our analysis shows that, looking at the behavior of the inverse graphical problem as data size increases, the methods exposed allow to rule out a wrong hypothesis.


# 1 Introduction

Concepts and tools from statistical mechanics turn out to be valuable resources to analyze the behavior of systems of very diverse nature, ranging from neuroscience [1, 2, 3] to systems biology [4, 5, 6], economics [7, 8, 9], finance [10, 11], sociology [12] and language evolution [13], just to name a few. In this work, we start from the use of statistical mechanics to characterize the behavior of complex optical systems [14, 15], in which many modes propagate or develop due to external sources and nonlinearities might be fundamental in describing the system behavior.

The study of Hamiltonian multi-body interaction systems as models to describe the interactions among electromagnetic modes in multimode lasers has provided important understanding on several experimental studies [16, 17, 18, 19]. In these works, the interests was mainly in the study and description of the electromagnetic output assuming some network of interactions among the modes. In this paper we will focus on statistical inference, i.e., on the inverse problem: the reconstruction of the statistical model parameters from data acquired in real experiments or in numerical simulations of the output.

Even though our original motivation arises in the framework of nonlinear optics and laser

physics [14, 20, 21, 22, 23, 24], in this work we will concentrate on state-of-the-art techniques to solve inverse problems. Indeed, the techniques here presented can be applied to a large class of models with multi-body interactions.

The most studied inverse problem within the statistical physics community is the inverse Ising problem. This can be regarded as the *E. coli* of statistical physics: the thermodynamic properties are known and the different techniques can be tested on the thermodynamic phases. In the Ising model the interacting variables are discrete spins, $\sigma = \pm 1$, while the couplings describing their interactions can be generically thought of as random variables chosen from desired probability distributions. Many inference techniques have been tested on the Inverse Ising model: from mean field methods [25] and their extensions [26, 27] to various likelihood maximization techniques [28, 29]. The literature on the inverse Ising problem is very wide, see e. g., Ref. [30] for a recent review. On the other hand, few studies can be found on continuous spin models [15, 31]. Moreover, inference techniques on systems with strong nonlinear responses have not yet been exhaustively explored. Beside the mentioned nonlinear optics, there are many other research fields in which nonlinear inference techniques are relevant: constraint satisfaction problems [32, 33], error correcting codes [34, 35], non linear neural networks [36], peptide sequences in DNA [37], non linear electric networks [38], fish shoals [39, 40], heterogeneous and frustrated glassy systems [41, 42, 43, 44, 45, 46, 47, 48, 49].

This study follows a previous paper [50], where we have presented the physical systems of interests as well as the inference techniques used, namely *pseudolikelihood methods* with $\ell_1$-*regularization* and *decimation*. Pseudolikelihood methods have proved to be of primary importance in a variety of research areas, e.g., in neuroscience for investigating populations of neurons [51], in the reconstruction of gene regulatory networks [52], in the determination of protein structure [4, 53, 54]. In the present work, we extend the analysis to provide a deeper understanding and a broader outlook and perspective on the problem and on the results obtained. In particular, i) we explain more in details the methods proposed in [50] to predict the $\lambda$ regularizer for the $\ell_1$-*regularization* in order to restrict ourselves in a minimum reconstruction error regime; ii) we clarify the criterion chosen for the halt of the decimation procedure; iii) we compare the dynamics realized on the inferred network of interactions with the real one, iv) we analyze the distributions of the inferred interaction couplings in relation to different underlying thermodynamic phases, v) we present also the results obtained starting from a wrong hypothesis: while the real dynamics is the one of a system characterized by a nonlinear response, the Hamiltonian is assumed to contain only 2-body interaction terms. In a previous paper [15], we compared the performances of the pseudolikelihood maximization (PLM) estimator with other estimators based on mean field approximations. With the pseudolikelihood maximization we obtained better performances even in the low sampling regime. In this work we, thus, concentrate on techniques based on the pseudolikelihood maximization. We note that studying systems with strong non-linearities, the mean field techniques would have been much more computational demanding with respect to the PLM.

The present paper is developed along the following scheme: in Sec. 2 we quickly introduce the models as well as the physical problems of interest; in Sec. 3 we explain in details the inference techniques that have been adopted; in Sec. 4 we present the results obtained. In section 5 we look at the outcomes of statistical inference starting from a wrong hypothesis; in Sec. 6 conclusions and further perspectives are elaborated.

## 2 Test Model

In this section the models and the physical systems of interest are introduced. The interested reader can find more details in [14]. This paper is organized in such a way to let the reader interested only in inference techniques applied to nonlinear systems to skip this section.

### 2.1 Complex Spherical Model: Relevance in Photonics

The class of Hamiltonians that we will consider has the form

$$\mathcal{H} = -\frac{1}{8} \sum_{jklm}^{\text{d.i.}} J_{jklm} \, a_j a_k^* a_l a_m^* + \text{c.c.} \tag{1}$$

where, in the most general case, $a_j$ are complex numbers and $J_{ijkl}$ are the complex interaction couplings among them. The sum considered in Eq. (1) is subjected only to distinct indices. A complete derivation of the above model in the context of optics is presented in Ref. [21]. The Hamiltonian description comes about as the electromagnetic field can be expressed as

$$\tilde{E}(\boldsymbol{r}, t) = \sum_k a_k(t) \boldsymbol{E}_k(\boldsymbol{r}) e^{\iota \omega_k t} + \text{c.c.} \tag{2}$$

where $\boldsymbol{E}_k(\boldsymbol{r})$ are the time-independent solutions of the wave equation in the medium, the so-called normal modes. The $a_k(t)$'s vary on a time scale much longer than $\omega_k^{-1}$ [55]. Each mode can then been seen as a phasor "spin", complemented with its own mode intensity $A_k^2 = |a_k|^2$ and phase $\phi_k = \arg(a_k) \in [0, 2\pi]$. As we can see from Eq. (1), we consider systems with 4-body interaction terms, representing the first nonlinear order term for optical systems with time reversal symmetry (i. e., with optical susceptibility $\chi^{(3)}$ ). Eventually, because of the total power constraint and regulatory mechanisms such as, e.g., gain saturation responsible for the stationarity of the laser regime, the $a_k$'s satisfy a global spherical constraint, i. e., $\sum_{k=1}^{N} |a_k|^2 = \text{const} \times N$, which assures a bounding of the energy (1). The inverse of the pumping rate plays the role of the effective temperature. The model introduced in Eq. (1) can, then, describe how the intensity is distributed among the modes in the stationary regime and how and if the phases of the modes would be synchronous. In this setup, the interaction couplings $J_{jklm}$ express the interaction among the modes due to the competition for the energy in the same region of the gain medium.

Starting with Ref. [56], Gordon and Fischer were the first to consider multimode lasers in a statistical-mechanical framework. They studied the statistical properties in homogeneous cavities taking into account non-linear effects like gain saturation and intensity dependent refractive index: the system shows a thermodynamic phase transition, i.e., the transition to the so-called multi-mode mode-locking ultrafast laser regime, in which the modes oscillate in a phase-locked behavior. Different thermodynamic phases are also observed in random lasers [20, 21, 57, 58].

#### 2.1.1 XY model, *aka Quenched Amplitude* model

Being interested in studying possible mode-locking regimes characterized by strong correlations among the phases of the modes, one can consider to investigate the situation of all the intensities $|A_j|$ as being *quenched* with respect to the phases, i.e., varying on much longer timescales with respect to the phases and considered as constants. Starting from Eq. (1), by a rescaling of the coupling coefficients $A_j A_k A_l A_m J_{jklm} \to J_{jklm}$, we are left with the Hamiltonian

of the $XY$ model:

$$
\begin{aligned}
\mathcal{H} \;=\; & -\frac{1}{8} \sum_{jklm}^{\text{d.i.}} \big[ J_{jklm}^{R} \; \cos(\phi_j - \phi_k + \phi_l - \phi_m) \\
& + \; J_{jklm}^{I} \; \sin(\phi_j - \phi_k + \phi_l - \phi_m) \big],
\end{aligned}
\tag{3}
$$

being $J^R$ and $J^I$ the real and imaginary part of the network coupling. We have analyzed the inverse problem for both models: the complex spherical/phasor model, Eq. (1), and the $XY$/rotor model, Eq. (3). Introducing the $XY$ model gives also the possibility to analyze the dynamics on sparse graphs with a number of total quadruplets scaling only with $N$. Indeed, if the number of total quadruplets does not scale at least with $N^2$, the dynamics of the spherical model displays power condensation: all the energy condensates in a quadruplet of phasors, the behavior is not homogeneous and ergodicity is not satisfied.

## 2.2 Frequency Matching Condition and mode-locking

Deriving Eq. (1) (see, for example, [14]), one can see that 4-modes can interact if and only if the following Frequency Matching Condition (FMC) is satisfied, i.e.,

$$
|\omega_j - \omega_k + \omega_l - \omega_m| \lesssim \gamma,
\tag{4}
$$

where with $\omega_k$ we indicate the frequency of mode $k$ and with $\gamma$ the linewidth. Our interest will rely on whether or not the inference techniques proposed are able to reconstruct this underlying structure. Moreover, knowing this constraint in advance, it might be possible to determine the frequency distribution of the modes. The 24 possible permutations of the 4 indexes in Eq. (4) can be divided into 3 non-equivalent subsets of 8 permutations each. So, the FMC can be satisfied by one, or more, of these independent classes of permutations. The term "narrow band" [56, 58, 59, 60, 61] indicates the case in which all modes oscillate in a relatively small frequency bandwidth, i.e., $\omega_l \simeq \omega_0$ within the linewidth $\gamma$ of mode $l$ and the FMC does not play any-role in the system behavior. For comb-like distributed frequencies, we have $\omega_l = \omega_0 + l\delta$ with $\gamma \ll \delta$. In this case, the systems is not fully connected and the connectivity of each mode depends on its frequency: the FMC plays a role in the construction of the interaction network. These graphs are termed Mode-Locked (ML) graphs. In this paper, we will consider both narrow band and ML graphs.

# 3 Inverse problem: data and inference techniques

The aim of supervised statistical learning is to predict the model parameters from data describing the dynamics of the system. Having modeled our optical system in terms of complex spherical or $XY$ spins interacting on a given graph, we now want to analyze the inverse problem: reconstructing the network of interaction as well as the coupling strength.

Since at present we have no access to experimental data, we will use numerical experiments to generate data providing, as a first step, an easy interface with real world experiments. The data used have been generated through Monte Carlo numerical simulations of the systems at equilibrium. Both systems, Eq. (1) and Eq. (3) have been simulated. We have considered both the case of sparse graphs, in which the number of interacting quadruplets scales like the number of variables, $N_q \propto N$, and more dense graphs, in which $N_q \propto N^3$. Notice that a complete dense graph would contain $O(N^4)$ interacting quadruplets. Furthermore, we have considered strict frequency matching conditions, cf. Eq. (4), based on comb-like single mode resonance distributions ($\gamma \ll \delta\omega$), as well as *narrow-band* conditions ($\gamma > \delta\omega$). Considering

the physical model of interests, the couplings $J$s will depend on the spatial distribution of the modes and on the non-linear response of the system. Particularly for random lasers, in which the modes are randomly distributed in space and the nonlinear susceptibility is highly inhomogeneous, we expect a variation of the value of the coupling among the interacting quadruplets. Because of the partial knowledge of modes localization and the very poor knowledge of the nonlinear response so far in experiments, the random values for the $J$s can be taken from any physically reasonable arbitrary probability distribution. We will then take the couplings of a multibody interacting network, with number of variables $N$ and number of couplings $N_q \sim N^z$, as generated either through a bimodal distribution, i.e.,

$$P(J) = 1/2[\delta(J - \hat{J}) + \delta(J + \hat{J})], \tag{5}$$

with $\hat{J} = 1/N^{(z-1)/2}$, or through a Gaussian distribution of mean square displacement $\sigma \sim \hat{J}$. Indeed, we can analyze the performance of the inference techniques for both discrete and continuously distributed couplings. The methods exposed also work for the simpler cases of uniform couplings like in standard mode-locking lasers [22, 56, 62, 63]. We inferred data within the equilibrium hypothesis, expressed by the Boltzmann-Gibbs distribution in the likelihood function. To reduce the Monte-Carlo steps required for thermalization we used the parallel tempering algorithm.

For the "narrow band" approximation, where the frequencies of the modes do not play any role in the construction of the interaction graph, we generate instances of Erdös Rényi (ER) graphs: each quadruplet is added to the graph independently, with probability $M/\binom{N}{k}$, where $M$ is the total number of quadruplets in the graph. In order to obtain a Mode-Locked (ML) graph, starting from a so generated ER graph, we remove those quadruplets that do not satisfy the Frequency Matching Condition (FMC), Eq. (4). As proved in Appendix A, in the thermodynamic limit, the number of removed quadruplets scales like $N/2$ while the node connectivity distribution tends to a Poissonian, as in ER-like graphs [64]. On the other hand, finite size effects are clearly stronger in the ML graph with respect to the ER-like graph for the relatively small simulated sizes.

## 3.1 Pseudolikelihood method

A standard approach used in statistical inference is to predict the model parameters by maximizing the likelihood function. This technique, however, requires the evaluation of the partition function that, in the most general case, concerns a number of computations scaling exponentially with the system size. Different approaches have then been used: Boltzmann machine learning [65, 66], mean field methods [67, 68] with various extensions [26, 27, 69, 70]. A local alternative to the likelihood function was introduced and referred to as Pseudo Likelihood Function (PLF) [28]. It was first developed for spatial models in Ref. [71] and later extended as an alternative to maximum likelihood function for networks. The most attractive part of the PLF is its computational tractability in comparison to the likelihood function. It keeps a good balance between the computational complexity and the efficiency of the estimation. The PLF is maximized with respect to its parameters to find the corresponding estimators. This method is known as Pseudo Likelihood Maximization (PLM). Such a logistic regression based method proves to work very efficiently on sparse networks. As well as the likelihood maximum estimator, the PLM estimator is consistent and asymptotically normal , i.e., as the number of training samples increases (i) the inferred values tend to the true values and (ii) the distribution of the inferred parameters tends to a Gaussian one. We are now deriving the PLF for non-linearly interacting wave systems.

Using Eq. (1), we have that the probability of observing a configuration $\boldsymbol{a}$ given a set of

couplings $J$ is:

$$P(\boldsymbol{a}|\boldsymbol{J}) = \frac{1}{Z[\boldsymbol{J}]} \exp\left\{-\beta\mathcal{H}[\boldsymbol{a}|\boldsymbol{J}]\right\}. \tag{6}$$

Eq. (1) can be rewritten as:

$$\mathcal{H}[\boldsymbol{a}] = -\frac{1}{8}\sum_{j=1}^{N} a_j H_j[\boldsymbol{a}_{\backslash j}] + \text{c.c.} \tag{7}$$

Where $a_{\backslash j}$ is the set of all amplitudes but $a_j$ and we have defined the complex-valued local effective fields as

$$H_j[\boldsymbol{a}_{\backslash j}] = \frac{1}{4}\sum_{klm\neq j}^{\text{d.i.}} J_{jklm}\mathcal{F}_{klm} \tag{8}$$

$$\mathcal{F}_{klm} = \frac{1}{3}\left[a_k^* a_l a_m^* + a_k a_l^* a_m^* + a_k^* a_l^* a_m\right]. \tag{9}$$

We want to determine $P_i(a_i|\boldsymbol{J},\boldsymbol{a}_{\backslash i})$ that, using Eq. (7), can be written as

$$P_i(a_i|\boldsymbol{J},\boldsymbol{a}_{\backslash i}) = \frac{\prod_{j=1}^{N}\exp\left\{\frac{\beta}{8}a_j H_j[\boldsymbol{a}_{\backslash j}]\right\}}{\sum_{\{a_i\}}\prod_{j=1}^{N}\exp\left\{\frac{\beta}{8}a_j H_j[\boldsymbol{a}_{\backslash j}]\right\}}. \tag{10}$$

All the terms in $\sum_j a_j H_j[\boldsymbol{a}_{\backslash j}]$ that do not depend on $a_i$ will simplify with the denominator. We write explicitly:

$$\sum_j a_j H_j[\boldsymbol{a}_{\backslash j}] = a_i H_i[\boldsymbol{a}_{\backslash i}] + \sum_{j\neq i} a_j H_j[\boldsymbol{a}_{\backslash j}]. \tag{11}$$

The sum over $j$ of the terms that depend on $a_i$ gives another term $a_i H_i[\boldsymbol{a}_{\backslash i}]$ and

$$P_i(a_i|\boldsymbol{a}_{\backslash i}) = \frac{\exp^{\left(\frac{\beta}{4}a_i H_i[\boldsymbol{a}_{\backslash i}]+\text{c.c.}\right)}}{Z_i[\boldsymbol{a}_{\backslash i}]}, \tag{12}$$

where

$$Z_i[\boldsymbol{a}_{\backslash i}] \equiv \sum_{\{a_i\}}\exp^{\left(\frac{\beta}{4}a_i H_i[\boldsymbol{a}_{\backslash i}]+\text{c.c.}\right)}.$$

The factor 4 will be absorbed in the definition of inverse temperature. The integral sum over $a_i$ can be successfully carried out by evoking the global *spherical* constraint $\sum_j |a_j|^2 = \epsilon N$, with constant $\epsilon$. Given all the $\boldsymbol{a}_{\backslash i}$, indeed, the value of $|a_i|$ is fixed by

$$|a_i| = \sqrt{\epsilon N - \sum_{j\neq i} |a_j|^2} \tag{13}$$

and $\sum_{a_i}$ simply reduces to an integral on the angular phase variable $\phi_i \in [0:2\pi[$.

Assuming that we are given $M$ *independent* configurations $a^\mu$, $\mu \in 1,\dots,M$, extracted from the Gibbs measure, the log-pseudolikelihood function eventually reads

$$\mathcal{L}_i = \sum_{\mu=1}^{M}\beta\left(a_i^\mu H_i[\boldsymbol{a}_{\backslash i}^\mu]+\text{c.c.}\right) - \sum_{\mu=1}^{M}\ln Z_i[\boldsymbol{a}_{\backslash i}^\mu]. \tag{14}$$

Next step is to minimize $-\mathcal{L}_i$ with respect to the Hamiltonian parameters that we want to infer: $\{J\}$. The stationary solution, in general, can only be computed by using a local gradient-based minimization [72]. To do so we need to compute explicitly the partial derivatives of $-\mathcal{L}_i$ with respect to each coupling constant:

$$\frac{\partial(-\mathcal{L}_i)}{\partial J_{ijkl}} = \sum_{\mu=1}^{M} \mathcal{F}_{jkl}^{\mu} \left[ \langle a_i \rangle_i^{\mu} - a_i^{\mu} \right], \tag{15}$$

where we denoted

$$\langle(\ldots)\rangle_i^{\mu} \equiv \frac{1}{Z_i[\boldsymbol{a}_{\backslash i}^{\mu}]} \sum_{\{a_i\}} (\ldots) \exp\left\{ \beta a_i H_i[\boldsymbol{a}_{\backslash i}^{\mu}] + \text{c.c.} \right\}. \tag{16}$$

### 3.1.1 Pseudolikelihood functional with rotor variables

Rewriting the complex amplitude in polar coordinates $a_i = A_i e^{\iota \phi_i}$ we have the following expression for the marginal, Eq. (12),

$$
\begin{aligned}
P_i(A_i, \phi_i | \boldsymbol{A}_{\backslash i}, \boldsymbol{\phi}_{\backslash i}) &= \frac{\exp\left\{ \beta A_i \left[ H_i^R \cos\phi_i + H_i^I \sin\phi_i \right] \right\}}{Z_i[\boldsymbol{A}_{\backslash i}, \boldsymbol{\phi}_{\backslash i}]} \\
&= \frac{\exp\left\{ \beta A_i |H_i| \cos(\phi_i - \gamma_i) \right\}}{2\pi \int dA_i \, I_0(\beta A_i |H_i|)},
\end{aligned} \tag{17}
$$

where

$$|H_i| = \sqrt{\left(H_i^R\right)^2 + \left(H_i^I\right)^2}, \tag{18}$$

$$\gamma_i = \arctan\frac{H_i^I}{H_i^R}, \tag{19}$$

and $I_0(x)$ is the modified Bessel function of the first kind:

$$I_0(x) = \frac{1}{2\pi} \int_0^{2\pi} d\vartheta \, e^{x \cos\vartheta} d\vartheta.$$

When the couplings are considered real-valued, the polar expressions of the local effective fields in Eq. (7) can be rewritten by substituting Eq. (9) with

$$
\begin{aligned}
\mathcal{F}_{jkl}^R &= \cos\phi_j \cos\phi_k \cos\phi_l \\
&+ \frac{\cos\phi_j \sin\phi_l \sin\phi_k + \cos\phi_l \sin\phi_j \sin\phi_k + \cos\phi_k \sin\phi_j \sin\phi_l}{3}
\end{aligned} \tag{20}
$$

$$
\begin{aligned}
\mathcal{F}_{jkl}^I &= \sin\phi_j \sin\phi_k \sin\phi_l \\
&+ \frac{\sin\phi_j \cos\phi_l \cos\phi_k + \sin\phi_l \cos\phi_j \cos\phi_k + \sin\phi_k \cos\phi_j \cos\phi_l}{3}.
\end{aligned} \tag{21}
$$

In this case, the log-pseudolikelihood functional $\mathcal{L}_i$, Eq. (14), and its gradient, Eq. (15), simplify to

$$-\mathcal{L}_i = \sum_{\mu=1}^{M} \left\{ \ln 2\pi I_0 \left( \beta \left| H_i(\boldsymbol{\phi}_{\backslash i}^{\mu}) \right| \right) - \beta \left[ H_i^R(\boldsymbol{\phi}_{\backslash i}^{\mu}) \cos\phi_i^{\mu} + H_i^I(\boldsymbol{\phi}_{\backslash i}^{\mu}) \sin\phi_i^{\mu} \right] \right\} \tag{22}$$

$$\frac{\partial(-\mathcal{L}_i)}{\partial J_{ijkl}} = \sum_{\mu=1}^{M} \left\{ \mathcal{F}_{jkl}^{\mu} \left[ \frac{I_1(\beta |H_i(\boldsymbol{\phi}_{\backslash i}^{\mu})|)}{I_0(\beta |H_i(\boldsymbol{\phi}_{\backslash i}^{\mu})|)} \frac{H_i(\boldsymbol{\phi}_{\backslash i}^{\mu})}{|H_i(\boldsymbol{\phi}_{\backslash i}^{\mu})|} - e^{\iota \phi_i^{\mu}} \right] + \text{c.c.} \right\}. \tag{23}$$

We note that in the mean-field-like inference, one crucial minimal criterion for the inverse problem to be tractable is that the number of data observations $M$ has to be larger or equal to $N$, in order for the correlation matrix to be invertible. In the present method this lower bound is not strictly requested, though a unique solution to PLM is guaranteed only when $M$ is larger than the number of coupling constants to be inferred. We will now compare two pseudo-likelihood techniques: PLM with $\ell_1$-regularization [28, 73] and the PLM with decimation [29].

## 3.2 Improved Pseudo Likelihood Maximization with $l_1$ regularization: hypothesis testing

A regularizer is usually required to prevent overfitting in the minimization procedure. It is done so by putting a kind of prior to enforce couplings to take small values [28]. One particular kind of regularizer used is the $l_p$-*norm regularizer*. It is defined for a vector $x = (x_1, \ldots, x_N)$ as,

$$||x||_p = \sqrt[p]{|x_1|^p + \cdots + |x_N|^p}. \tag{24}$$

Any regularizer should be convex so that the convexity of the inverse problem remains intact. In this approach, we add an $\ell_1$ norm:

$$\mathcal{L}_i \to \mathcal{L}_i - \lambda_J \sum_{jklm}^{\text{d.i.}} |J_{jklm}|. \tag{25}$$

The $\ell_1$ has proved to be special with respect to $p > 1$ norms, because it performs well on sparse problems, where only a few parameters are actually non-zero. This is the case, e. g., of sparse graphs, in which the number of couplings per variable, the "connectivity" $c = N_q/N$, does not grow with $N$[1]. The reconstruction of the topology is further enhanced by the so called $\delta$-thresholding [73], i.e., couplings that are inferred less that $\delta$ are set to zero. This technique comes with its own shortcoming in the decision of the value of $\delta$. Indeed, there can be cases where the zero and non-zero couplings are not clearly separated [29], see, e.g., the left panel of Fig. 12 in Sec. 4.4 for low number of samples $M$.

With the knowledge of the probability distribution of the estimators, this problem can be overcome by using an accurate hypothesis testing scheme. It is known that as $M \to \infty$, the probability distribution of the PLM estimator converges to a Gaussian distribution centered around the true value of the coupling with variance estimated by the diagonal elements of the inverse of the Fisher information matrix [75]. The elements $\mathcal{I}_{ab}^i$ of the information matrix are defined through:

$$\mathcal{I}_{ab}^i = -\frac{\partial^2 \mathcal{L}_i}{\partial J_a \partial J_b}\Big|_{\hat{\mathbf{J}}}, \tag{26}$$

where $a, b$ indicate two possible quadruplets to which node $i$ might belong to, i.e., $\mathcal{I}_{ab}^i \equiv \mathcal{I}_{jkl,j'k'l'}^i$. Then, we can determine, for every estimated value, if it is compatible with a Gaussian centered in zero, i.e., if the hypothesis for the true coupling to be zero must be accepted or rejected. Note that, in order for the procedure to be consistent, the eigenvalues of the Fisher Information Matrix need to be bounded from below. This condition together with the requirement that the entries of $\mathcal{I}_{ab}^i$ related to nonneighbors of $i$ cannot exercise an overly strong effect on the subset related to the neighbors of $i$ assures that the PLM with $l_1$ regularization has a unique

---

[1] One might argue that in dense graphs where $c \sim N^\alpha$, with $\alpha > 0$ (see Appendix B), the $\ell_1$-regularization for sparse models is ineffective. However, though this regularization does not bring any advantage with respect to $\ell_{p>1}$ regularizations in absence of sparsity, it does not hinder PLM either. This suites the so-called "bet on sparsity" principle [74]: "use a procedure that does well in sparse problems, since no procedure does well in dense problems".

solution and correctly reconstruct the neighbors of $i$ if enough number of samples are provided ($M$ larger than the number of parameters to be inferred) [28]. These requirements are then checked[2] in our data.

The hypothesis testing is subsequently developed as follows. At the initial step, after finding the maximum of the PLF, the inverse of the Fisher information matrix is evaluated, Eq. (26). In the case of rotors (but proceeding analogously for phasors) from Eq. (22), we have:

$$
\mathcal{I}_{ab}^{i} = \sum_{\mu=1}^{M} \mathcal{F}_a^{\mu} \mathcal{F}_b^{\mu}
$$

$$
\left\{ \left( \frac{H_i(\boldsymbol{\phi}_{\backslash i}^{\mu})}{\left| H_i(\boldsymbol{\phi}_{\backslash i}^{\mu}) \right|} \right)^2 \mathcal{B}\left( \left| H_i(\boldsymbol{\phi}_{\backslash i}^{\mu}) \right| \right) + \frac{I_1(|H_i(\boldsymbol{\phi}_{\backslash i}^{\mu})|)}{I_0(|H_i(\boldsymbol{\phi}_{\backslash i}^{\mu})|)} \frac{\left| H_i(\boldsymbol{\phi}_{\backslash i}^{\mu}) \right|^2 - \left( H_i(\boldsymbol{\phi}_{\backslash i}^{\mu}) \right)^2}{\left| H_i(\boldsymbol{\phi}_{\backslash i}^{\mu}) \right|^3} \right\}, \quad (27)
$$

$$
(28)
$$

where $\mathcal{B}(x)$ is defined as:

$$
\mathcal{B}(x) = \frac{1}{2} \left( \frac{I_2(x)}{I_0(x)} + 1 \right) - \left( \frac{I_1(x)}{I_0(x)} \right)^2.
$$

In Eq. (28), to simplify the notation, we have included the factor $\beta$ through a rescaling of the $J$s. The diagonal terms of the inverse Fisher matrix, $\hat{\sigma}_a$, are then computed as the estimators for the variances of the distributions $P(\hat{J}_a)$:

$$
\hat{\sigma}_a = \mathcal{I}_{aa}^{i}. \quad (29)
$$

As a further step, every coupling is hypothesized to be zero and it is verified whether every estimated value is compatible with the $P(J_a) = \mathcal{N}(0, \hat{\sigma}_a)$ distribution: we construct a confidence interval $C_n$ containing the estimated value $\hat{J}_a$ within a 97.5% probability; if the inferred $\hat{J}_a$ is contained in $C_n$ the zero hypothesis cannot be ruled out and that coupling is considered to be zero and taken away from the inferred network.

We conclude this section noting that, by maximizing each $\mathcal{L}_i$ separately, one has 4 different estimated values for each quadruplet coupling. For the final estimated value, the mean is usually evaluated but we remark that the information on the symmetry of couplings of the system is not used in the inference procedure. We will now see how, in the Pseudo Likelihood Maximization with Decimation, this problem is overcome, further reducing the number of unknown couplings by a factor four.

## 3.3 Pseudo Likelihood Maximization with Decimation

In the PLM with decimation [29], instead of maximizing $N$ different pseudo-likelihood functions $\mathcal{L}_i$, an average PLF $L$ over all variables is maximized [29]:

$$
L \quad \equiv \quad \sum_i \frac{1}{N} \mathcal{L}_i = \sum_{i=1}^{N} \frac{1}{N} \sum_{\mu=1}^{M} \frac{1}{M} \log P(a_i^{\mu} | a_{\backslash i}^{\mu}).
$$

$$
(30)
$$

One of the advantages over PLM-$l_1$ is that we are not perturbing the function to be maximized by adding a regularizing term and there is no choice of the $\lambda$ parameter to be carried out. To reconstruct the set of non-zero couplings, the smallest estimated couplings are recursively put

---

[2]The second can be checked after the reconstruction.

to zero and the maximization procedures is repeated until the best inferred model is achieved. Let us indicate with $J_1/J_0$ the set of non-zero/zero couplings. The procedure goes at follows. At the zeroth step of decimation, $J_0^*$, the inferred set of null couplings, is empty. At each step we maximize the $L$ function obtaining the set $J^*$ of inferred couplings. We sort them by absolute value and we move the least $\rho$ couplings from $J_1^*$ to $J_0^*$. That is, we *decimate* $\rho$ couplings from the system. We want to stop the decimation procedure when $J_0 = J_0^*$, i. e., all the couplings that are zero in the original system are put to zero in the inferred system. At each step, the PLF is between its value $\mathrm{PLF}_{\max}$, evaluated when all possible couplings are considered, i.e., a fully connected graph with $J_0^* = \emptyset$, and $\mathrm{PLF}_{\min}$, i.e., an empty graph with $J_1^* = \emptyset$:

$$\mathrm{PLF}_{\min} \quad \leq \quad \mathrm{PLF} \leq \mathrm{PLF}_{\max}. \tag{31}$$

When we decimate couplings that are actually in $J_0$, the value of PLF does not change from $\mathrm{PLF}_{\max}$ since we are decimating irrelevant couplings. As the set $J_0^*$ approaches the real $J_0$, the chance of eliminating an existing coupling increases and the PLF starts decreasing. To determine more precisely the fraction $x$ of remaining couplings where the PLF starts decreasing, a *tilted* PLF is defined:

$$t\mathrm{PLF} = \mathrm{PLF} - x\mathrm{PLF}_{\max} - (1-x)\mathrm{PLF}_{\min} \tag{32}$$

with

$$x = \frac{\text{non-decimated couplings}}{\text{total number of couplings}}. \tag{33}$$

At zeroth, when the graph is fully connected and $x = 1$, $t\mathrm{PLF}$ is zero. For an empty graph, $x = 0$ and $\mathrm{PLF}=\mathrm{PLF}_{\min}$, the $t\mathrm{PLF}= 0$ once again. As $x$ is decreasing from 1 to 0, we will observe first a linear increase up to a maximum and then a decrease [50]. The best representation of the real network occurs at the value of $x$ such that $t\mathrm{PLF}$ is maximum. The decimation stops there.

## 4    Multi-body inference results

### 4.1    $\ell_1$-regularization PLM and *a priori* $\lambda$ estimation: *no-match* parameter

For the PLM with $\ell_1$-regularization the value of the $\lambda$ regularizer is usually chosen arbitrarily and checked *a posteriori*: assuming that one knows the solution of the inverse problem for one set of data, the best $\lambda$ is the one yielding minimum reconstruction error for the inferred couplings and - possibly simultaneously - the best network reconstruction of the inferred system.

In this section, we develop a mechanism through which we can choose the best $\lambda$ regularizer *a priori*, without any knowledge of the couplings. Cross-Validation (CV) techniques are also often applied to determine the best value of $\lambda$ on supervised learning algorithms (see, e.g., Ref. [76]) and they do not require the knowledge of any solution of the inverse problem. A CV method might be developed on the following scheme: (i.) the observed configurations are divided in two sets, a training and a validating set; (ii.) the training set is used to fit the model, i.e., to determine the interaction couplings as a function of the trial value for the regularizer; (iii.) a Monte Carlo dynamics of the model with these inferred couplings is performed, the equilibrium configurations are acquired and the four point correlation functions, $C_{ijkl}^{\mathrm{mc}}$, are computed; (iv.) these correlation functions are, then, compared to those obtained from the configurations of the validating set, $C_{ijkl}^{\mathrm{val}}$; (v.) the optimal $\lambda$ is, finally, taken as the value that minimizes the distance among $C_{ijkl}^{\mathrm{mc}}$ and $C_{ijkl}^{\mathrm{val}}$.

CV techniques are quite computational demanding and the number of samples used to fit the model and infer the interaction couplings is further reduced because the method also requires a validating set of data. We will show the study and comparison of correlation functions in the original and the inferred systems in Sec. 4.3.3.

In this section, we venture into the system to find the optimal value of $\lambda$ from three different perspectives. We start considering a system with $N = 16$ spins on ER graph with number of quadruplets, $N_q = 32$. The analysis is shown at $T = 2.2$, next to the critical temperature $T_c \simeq 2.3$.

1. For the first perspective we evaluate the True Positive Rate (TPR), that is the fraction of true bonds appearing also in the inferred set of bonds, i.e., $J \in J_1 \cap J_1^*$, and the True Negative Rate (TNR), that is the fraction of missing bonds absent also in the inferred set of bonds, i.e., $J \in J_0 \cap J_0^*$. We, then, look at the minimum $\lambda$ for which the ratio TNR/TPR is equal to 1, i.e., the network is perfectly reconstructed. It is important to notice that the smaller the $\lambda$ the less perturbed is the PLF. The entire range of the ratio TNR/TPR vs. $\lambda$ is shown in Fig. 5.

2. The second perspective is the one which would give the minimum reconstruction error. To determine how far the inferred values $J_q^*$ of the distinct quadruplets $q \equiv \{i, j, k, l\}$ are from the true values $J_q$, we evaluate the reconstruction error:

$$\mathrm{err}_J \equiv \sqrt{\frac{\sum_q (J_q - J_q^*)^2}{\sum_q J_q^2}}. \tag{34}$$

In Fig. 7 we show the reconstruction error obtained for the $\lambda$ values show in Fig. 5.

3. For the third perspective, we introduce a new parameter called **no-match** parameter. Consider the inferred value, $J_q^*$, for the quadruplet $q \equiv \{i, j, k, l\}$. By maximizing each $\mathcal{L}_i$ separately, we have four different inferred values for $J_q^*$. The no-match parameter counts the quadruplets for which the result of the hypothesis testing was not the same for the four $J_q^*$s. Running the inference scheme for different values of $\lambda$ gives us different values of the no-match parameter. In Fig. 1 we plot the values of the no-match parameter as a function of $\lambda$. We see that as $\lambda$ increases, the no-match decreases. There is a value of $\lambda(M)$ beyond which the no-match parameter remains zero. We consider this as the optimal value of $\lambda$.

We stress that, in comparison with the requirements for the reconstruction error to be minimal or the TNR and TPR to be 1, to find $\lambda(M)$, for which the no-match parameter is first zero, we do not need any information about the real underlying network. We compare in Fig. 2 the performance of the newly introduced procedure 3 against the other two as a function of sample sizes $M$.

## 4.2 $\ell_1$-regularization PLM and estimators variance

Next, we move to the analysis of the variance of the inferred coupling distributions, evaluated as explained in Sec. 3.2 by computing the diagonal elements of the inverse of the Fisher information matrix $\mathcal{I}^i_{jkl,j'k'l'}$, cf. Eq. (26). We consider the same system as earlier, $N = 16$ XY spins on ER graph with $N_q = 32$. Fig. 3 shows the variance for each coupling estimator. Each 4-index set is labeled by an integer index of quadruplet $a \equiv \{i, j, k, l\}$, with $a = 1, \ldots, N(N-1)(N-2)(N-3)/24$. The quadruplet indices are arranged according to the ascending order of the coupling values: the left most are, thus, the $\sigma_a^2$'s associated with

non-zero couplings with negative value (12 of them in the considered system) and the right most are the $\sigma_a^2$'s associated with the non-zero quadruplets with positive value (20 of them). The rest in the middle, are the $\sigma_a$'s associated with zero couplings (1788 of them). No significative dependence on the index of quadruplet is observed. As expected from the consistence property of the PLM estimator, the $\sigma$ values decreases as the number of samples increases.

In Fig. 4, we show that average value of $\sigma$s as a function of temperature $T$. On top of the net decrease with increasing $M$, we observe that, when $T \sim T_c$, the $\sigma^2$ exhibit a sharp increase.

### 4.3  $\ell_1$-regularization and decimation PLM: a comparison

In this section, we compare the regularized and the decimated inference schemes using a variety of tests. For illustration, we choose a system constituted of $N = 16$ nodes.

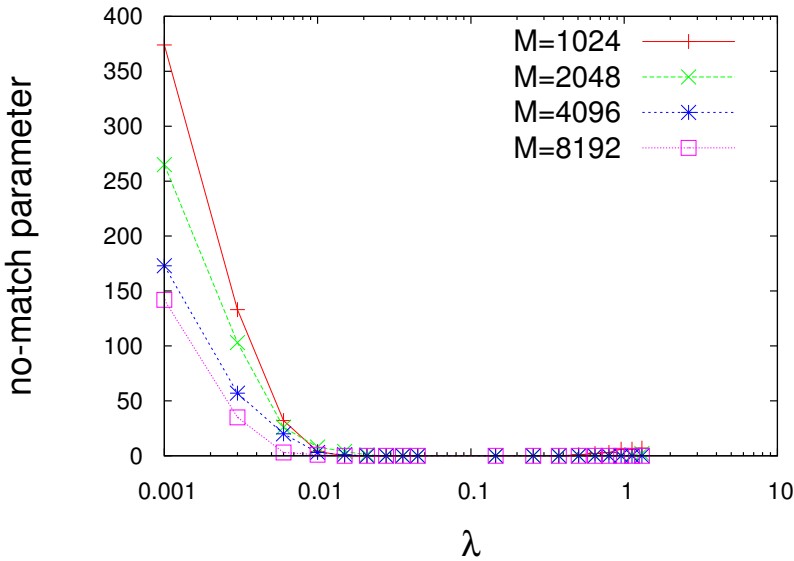

Figure 1: Value of no-match parameter obtained for various values of $\lambda$

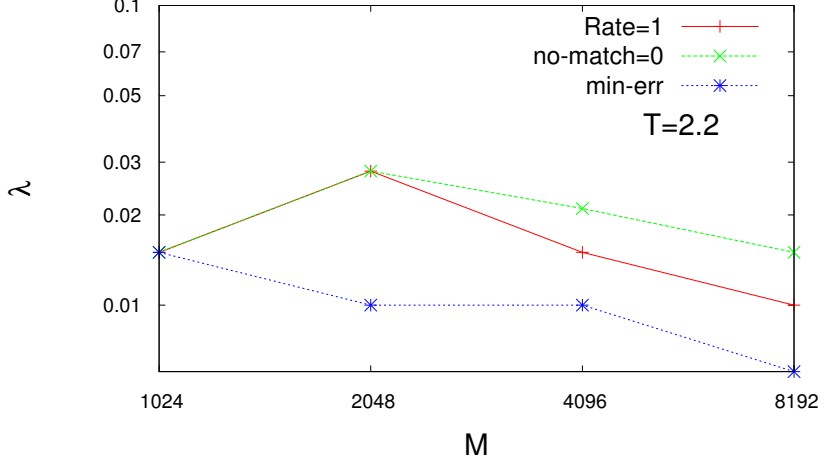

Figure 2: Best values of $\lambda$, for various data size $M$, obtained with different criteria: (i) TPR=TNR= 1, (ii) zero no-match parameter and (iii) minimal err$_J$.

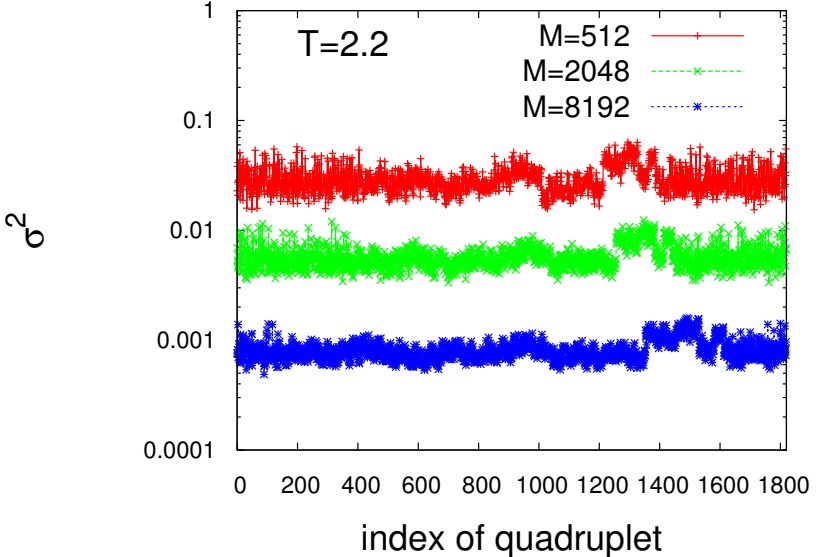

Figure 3: Variance $\sigma^2$ of the coupling estimator distribution for each quadruplet at various $M$ for $T = 2.2$. The system is constituted by $N = 16$ nodes, hence a total of 1820 quadruplets are displayed, sorted by increasing value of estimated $J$.

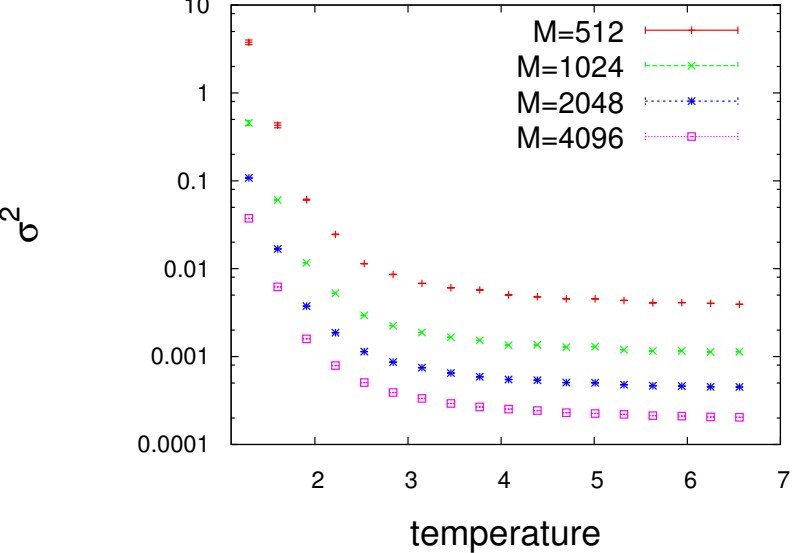

Figure 4: Average $\sigma^2$ for various $M$ as a fuction of $T$. For $T < T_c$, $\sigma^2$ show a strong increase.

### 4.3.1 True Positive and True Negative Rates

For the model with bimodal distributed couplings, in Fig.5, the ratio TNR/TPR is shown, both as a function of the $\lambda$ regularizer for $\ell_1$ and as a function of the number of non-decimated couplings $x$ for decimation. The top figure shows the behavior for $M = 1024$ at $T = 1.9$ as a function of $\lambda$. Together with $\ell_1$ we also use $\delta$-thresholding as previously reported. We used $\delta = 0.1, 0.01, 0.0001$. We see that the best result for the multi-decades range of $\lambda$ examined is given by the $\ell_1$ scheme with the hypothesis testing technique. For certain high values of $\lambda$, we see that the ratio becomes even larger than 1: with a strong regularization too many

couplings go to zero.

The bottom plot shows the results obtained with the PLM with decimation. $x = 1$ corresponds to the first step of decimation, TPR= 1 and TNR= 0. As we decimate couplings the TNR/TPR increases. At $x \sim 0.02$, TPR=TNR= 1. This would be the ideal point where to stop the decimation. In Fig. 6, we analyze the difference among $x^*$, the maximum point of the tPLF, Eq. (32), that is actually determined without any knowledge of the graph, and this ideal $x$. We can clearly see that the best results are obtained working around the critical temperature, here $T_c \sim 1.34$.

We can see that as $T$ depart from $T_c$ and $M$ is not large enough, $x^*$ departs from $x$. For the

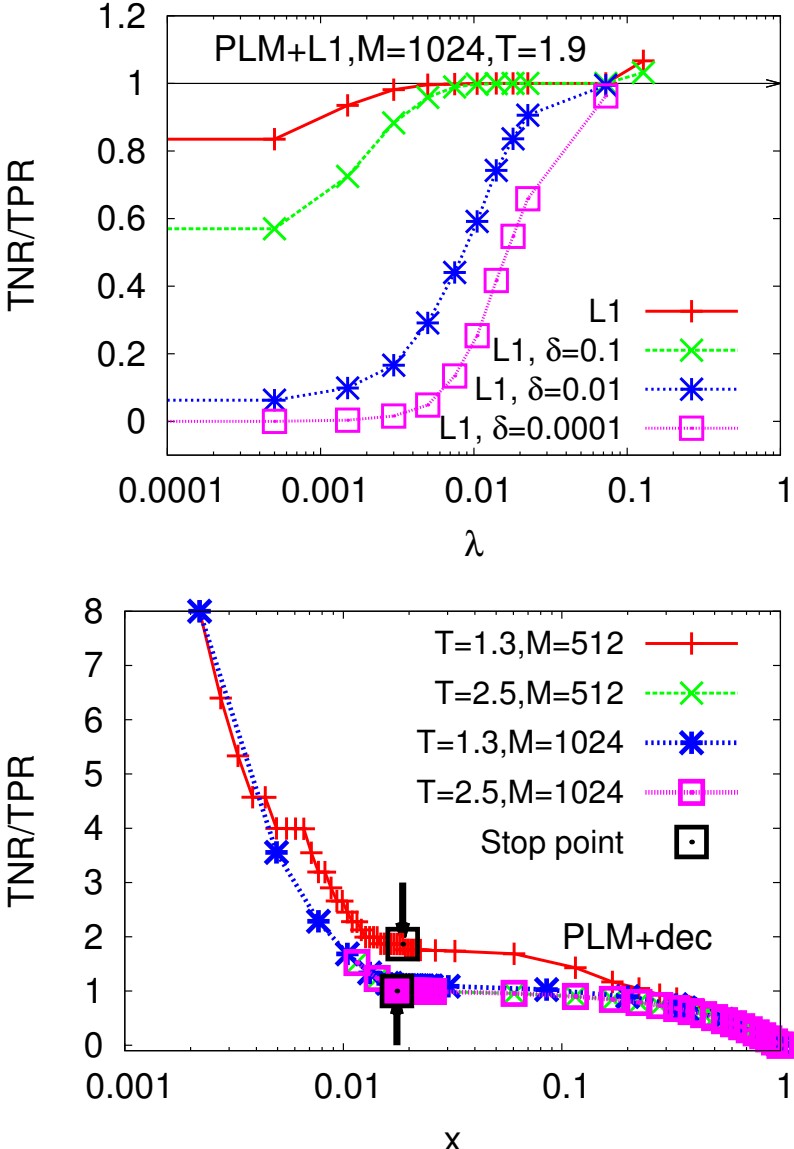

Figure 5: The TNR/TPR ratio vs. the regularizer $\lambda$ for the $\ell_1$-regularization PLM (top) and vs. the fraction $x$ of undecimated couplings for the decimation PLM (bottom). In the first case two different criteria are chosen to eliminate small bonds: with an *a-priori* threshold $\delta$ or by means of the *a posteriori* hypothesis testing procedure, indicated as $L1$ in the legend. Data are taken from the 4XY model on sparse, ER like graph, with $N = 16$, $N_q = 2N$. Here, $T_c \sim 1.34$

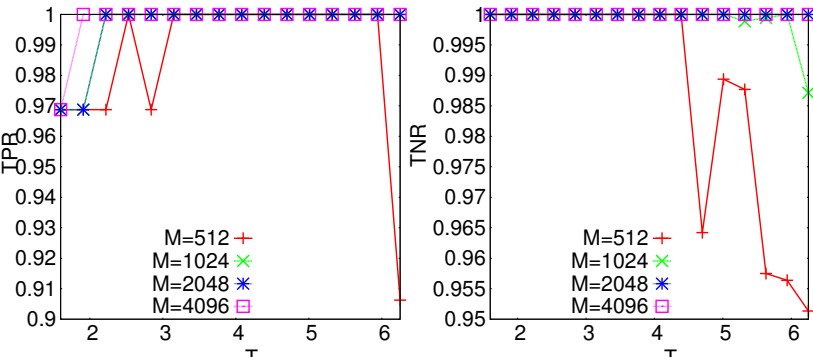

Figure 6: TPR (Left) and TNR (Right) for decimated networks at $x^*$, i.e., maximum point of the $t$PLF, varying $T$ and $M$ for the 4XY model on sparse, ML-like, graphs with $N_q = 47$, $N = 16$ $T_c(N = 16) \simeq 0.5$.

worst case, $M = 512$, we always have TNR<1. In Sec. 4.4.2, we will explain how we could reach better performances.

### 4.3.2 Reconstruction error

Fig. 7 shows the reconstruction error at $T = 2.2$ for increasing samples size. The error is plotted as a function of $\lambda$ for the $\ell_1$-regularization scheme and as a function of $x$ for the decimation scheme. In all the plots, we see that there is a dip in the error curve: there is a $\lambda$ value and a $x$ value for which we have minimum reconstruction error [3]. The results labeled by "L1" indicate that the zero couplings are selected through the $\ell_1$-regularization with the hypothesis testing scheme. In the decimation scheme the minimum is given at the ideal $x = 32/1820 \sim 0.18$ where also TPR = TNR = 1. In this case, this value coincides with the maximum of the $t$PLF.

In Fig. 8 we plot the error as a function of temperature, $T$, and sample size, $M$. On the left, we show the plot for $M = 4096$ for a wide range of temperatures ranging from $T = 0.5 < T_c$ to $T = 6.5 > T_c$. For the system under consideration, we have $N = 16$, $N_q = 2N$, bimodal disordered couplings and $T_c \simeq 2.3$. On the right, we plot the error as a function of $M$ ranging from $M = 512$ to $M = 8192$, for $T = 4.3$. We see also for this system the already established trends: $\mathrm{err}_J$ increases rapidly at low temperatures and decimation provides consistently less error than the $\ell_1$ scheme. Furthermore, the error scales as $1/\sqrt{M}$ for a given temperature.

### 4.3.3 Correlations

To get a better insight into the physical system that we are dealing with, we investigate the 4-point correlations, defined as

$$C_{ijkl} = \cos(\phi_i - \phi_j + \phi_k - \phi_l) \tag{35}$$

The scatter plot in Fig. 9 compares correlations obtained numerically simulating the dynamics of the original system and correlations obtained in a system whose coupling values are those inferred by pseudolikelihood maximization. We present cases for three different temperatures: $T = 0.8, 2.57$ and $7.03$, and, for three different sample size, $M = 512, 2048$ and $4096$. A green color reference line with slope 1 is drawn to compare with the optimal condition. For the high temperature case all correlations are zero, as expected in the paramagnetic phase.

---

[3]We plot the errors in the same plot because the range of $\lambda$ and $x$ is the same and they both show a dip in reconstruction error at nearby values. However, let us reinstate that $\lambda$ and $x$ are not related in any form. They are two different parameter for two different inference procedures.

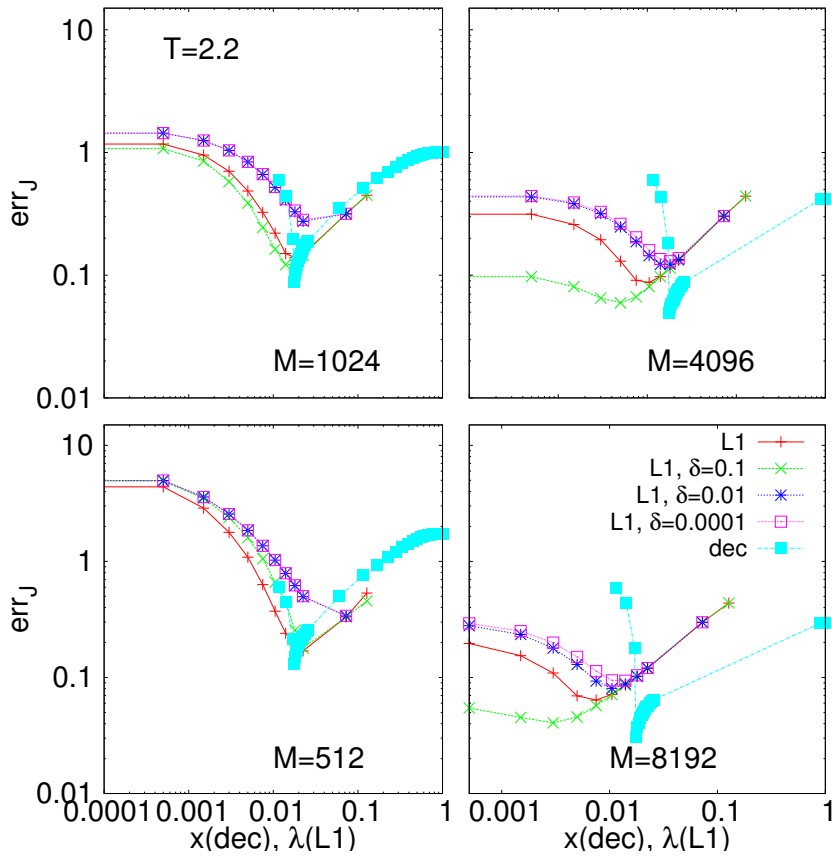

Figure 7: Reconstruction error vs $\lambda$ and $x$ for the 4XY model on sparse, Erdos-Renyi-like, graph with $N_q = N = 16$.

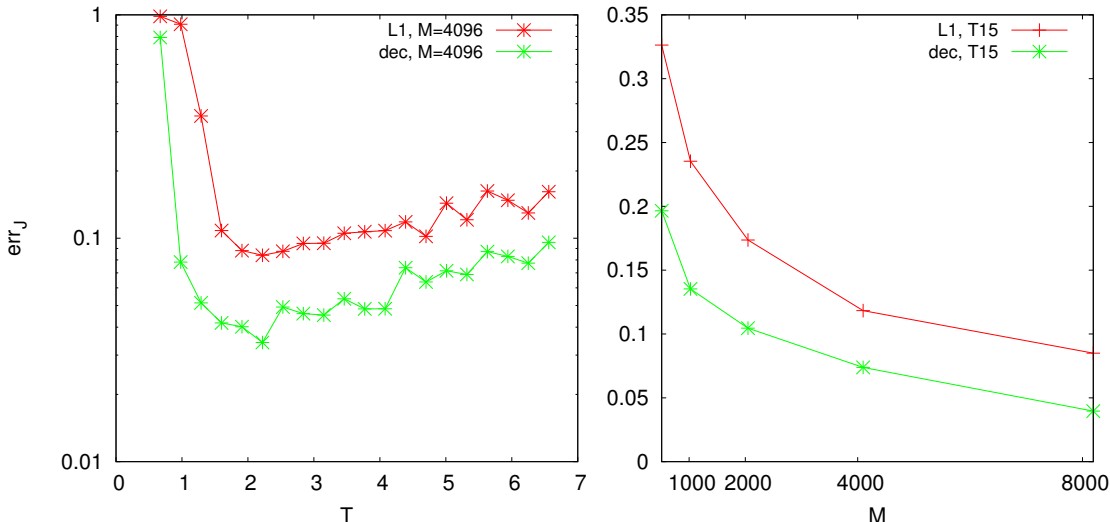

Figure 8: Reconstruction error versus $T$ for sample size M=4096 (left) and versus $M$ for temperature T=4.4 (right) for the 4XY model on sparse, Erdos-Renyi-like, graph with $N_q = N = 16$. The error obtained following both the $\ell_1$-regularized PLMs and the decimation PLM are shown.

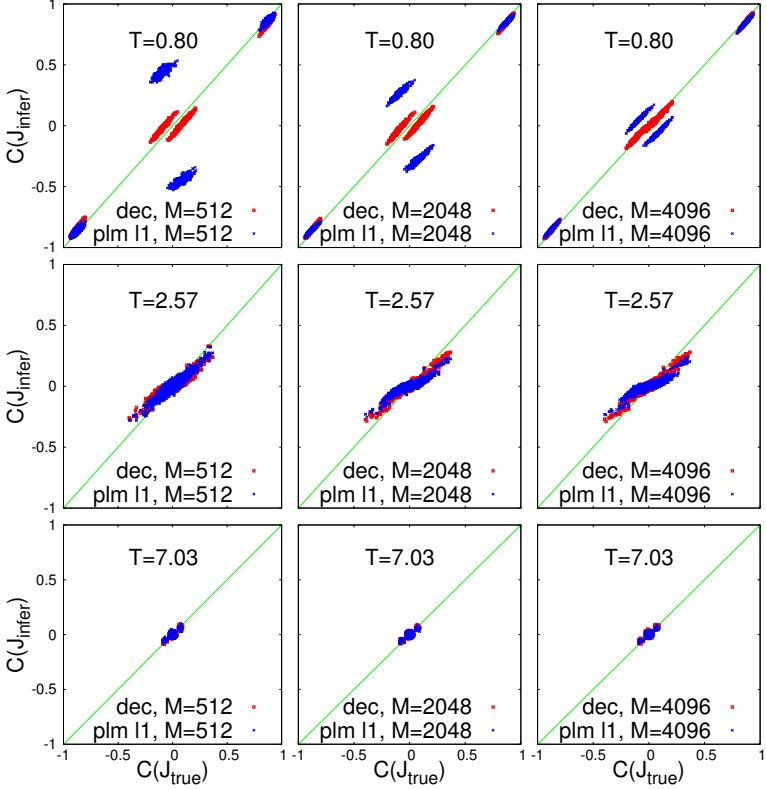

Figure 9: Comparison of the 4-point correlations computed in a Monte Carlo simulation of a system of $N = 16$ XY spins, $N_q = N$ with the original coupling network ($x$-axis, $J_{\text{true}}$) and with an inferred coupling network ($y$-axis, $J_{\text{inter}}$). Blue points are correlations measured on networks reconstructed by $\ell_1$-regularized PLM, red points corresponds to correlations on networks inferred by decimation PLM.

For $T \sim 2.6$, closer to $T_c \simeq 2.3$, the correlations depart from zero and for both the $\ell_1$-regularization and decimation schemes are spread along the reference line.

For even lower temperature, $T = 0.8$, we see that the correlations are separated into two groups: those distinctly different from zero and those close to zero. The decimation scheme yields correlation values nearby the reference line for small $M$ and with increasing $M$ data eventually collapse on the reference line. On the contrary, with $\ell_1$ regularization correlation points turn out to be distributed below and above the reference line but not along it. As $M$ increases the distance from the reference line tends to zero but much slower than with the decimation method.

## 4.4 Decimation results

We will now focus on the decimation procedure and display further results obtained through it.

### 4.4.1 Coupling values histogram

In Fig. 10, we plot the histogram of the inferred couplings for a system with $N = 32$ XY spins, $N_q = 192$ quadruplets on an ER graph. The couplings are generated randomly from a bimodal distribution. Three different sample sizes, $M = 8192$, 16384 and 65536, at $T = 3.3$ are considered. The first row shows the histograms as obtained at the zeroth step of

decimation. For low number of samples the distributions of the non-zero inferred couplings are not centered on the true values. As we increase $M$ this distance lowers. In Fig. 11, we show the histograms as obtained when the decimation procedure stops. In these cases, the network of interactions is correctly inferred. Moreover, the non-zero couplings are distributed around the true values for any $M$. As expected, by increasing $M$, the variance of the non-zero distributions decreases.

For the case of the complex spherical model, Eq. (1), we show an example of results in Fig. 12. Here the system is composed of $N = 32$ phasors with $N_q = 2360$, couplings follow a bimodal distribution. Results are at $T = 7.1 > T_c$. The figure on the left reports the couplings obtained at zeroth step of decimation. In this case for $M < 65536$ the distributions of zero and non-zero couplings overlap. When the decimation stops, figure on the right, we can see that the network of interactions is clearly reconstructed for $M > 8192$. For $M = 8192$ the algorithm is not able to correctly identify the zero couplings, i. e., TNR $< 1$: the distribution of the non-zero couplings cannot be clearly identified since a strong overlap with the distribution of zero couplings remains.

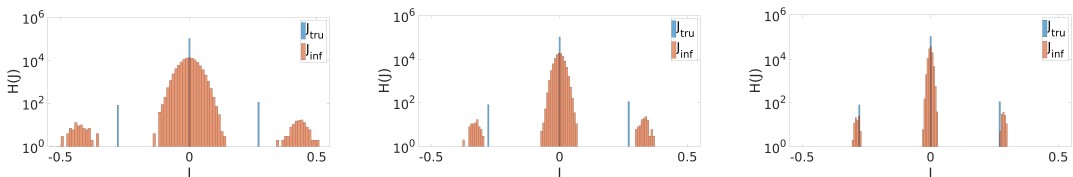

Figure 10: Histograms of the coupling constants $J$ at zeroth step of decimation. Here the system is composed of $N = 32$ XY spin; the original network has $N_q = 192$. Results are show at $T = 3.3$ for increasing sample size $M = 8192, 16384$ and $65536$ (from left to right).

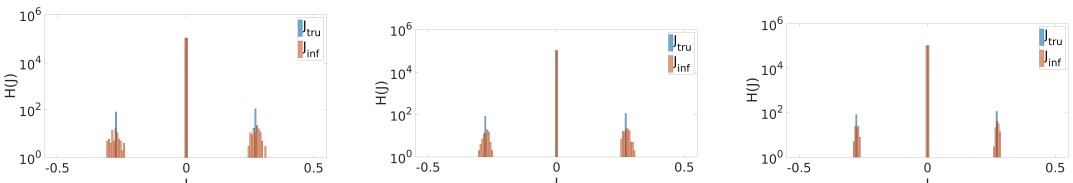

Figure 11: Histograms of the coupling constants $J$ as obtained when decimation stops for the system of Fig. 10.

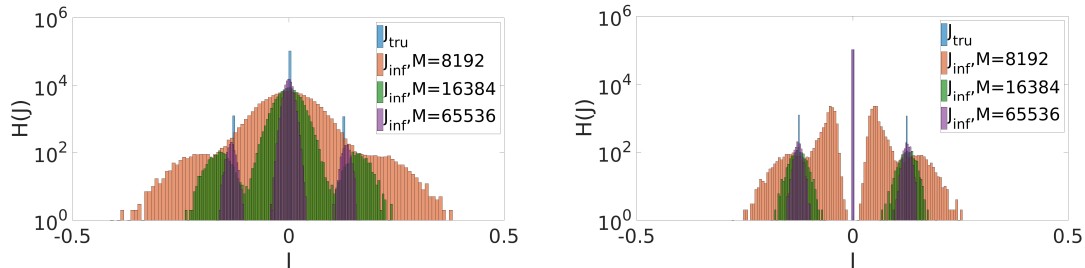

Figure 12: (Left) Histograms of the coupling constants $J$ at zero step of decimation for the complex SM. The system is composed of $N = 32$ phasors and $N_q = 2360$. $T = 7.1 > T_c$. (Right) Histograms of the same system of Figure right at the stopping point of decimation. For $M = 8192$ the zero and non-zero distributions still overlap.

#### 4.4.2 Estimating decimation halt criterion

In PLM with decimation, the criterion to stop the decimation is to reach the maximum point of the tPLF. It has been reported earlier [15, 29, 50], however, that when $M$ is not enough or $T$ is too far from $T_c$, the peak of tPLF is not always sharp enough [50] to be unambiguously identified numerically. We have established an alternative halt criterion for these complicated cases. We consider the relative difference $\Delta_i$ in the tPLF as one passes from the network at decimation step $i$ to the one at step $i + 1$:

$$\Delta_i = \frac{tPLF_{i+1} - tPLF_i}{tPLF_i}. \tag{36}$$

In Fig. 13 we plot $\Delta_i$ as a function of $x$. During the decimation procedure, as we proceed further in the iteration, we find that there is a discontinuity in the $\Delta_i$ function. This discontinuity appears at the step at which a true coupling is decimated from the system. This point is chosen as new stopping point. The corresponding fraction of non decimated couplings is termed $x^\Delta$. This new stopping criterion comes with its own limitation: if there are not enough data samples available and we are in a very high temperature range, we cannot find any discontinuity in the $\Delta$ function.

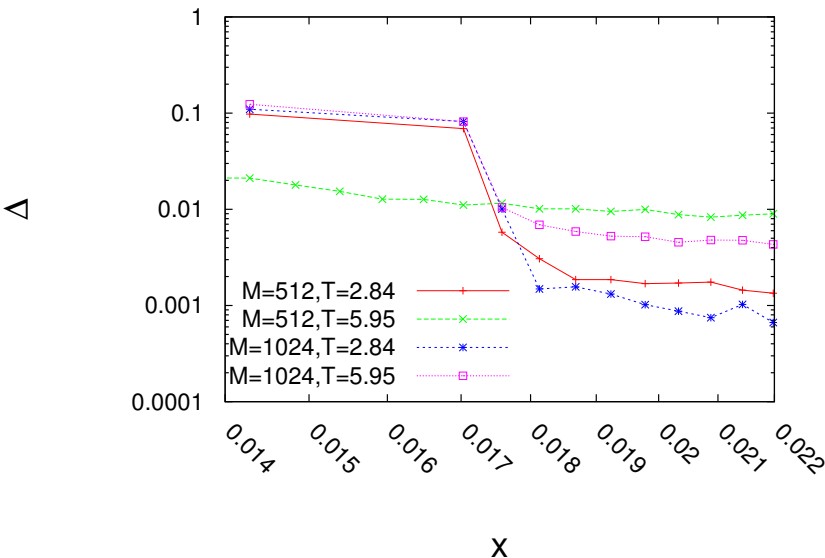

Figure 13: $\Delta$ vs $x$ for two different temperatures at low $M$ (512) and at a high $M$ (1024) in the 4-XY model. At low number of samples and high temperature, we cannot find a discontinuity in the $\Delta$ value.

In Fig. 14 we analyze the performance of this new criterion in respect to the maximum point of t$PLF$: $x^M$ indicates the maximum of tPLF and $x^m$ the minimum of the reconstruction error. $\alpha$ is defined as:

$$\alpha = \frac{M}{N_q}. \tag{37}$$

In Fig. 15, we plot these results for a system of $N = 32$ spins: in this case we are far from $T_c$ and the new criterion outperforms the previous one also for small $\alpha$.

In Fig. 16, we show how the position $x^M$ of the tPLF peak varies varying $T$ and $M$. The discontinuity of $\Delta$, on the other hand, though it smoothens and eventually disappears as $T$ increases, always stays at the same $x^\Delta$ value when it is observabel. In the figure we consider two different temperatures, $T = 2.84$ and 5.95 both bigger than $T_c$.

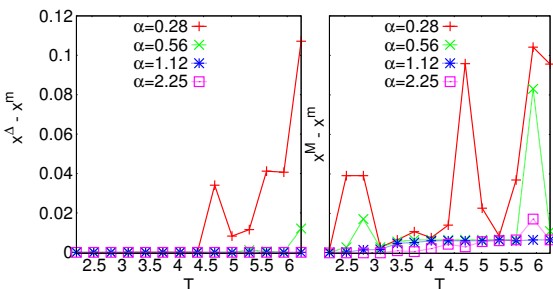

Figure 14: Left: Plot of $x^{\Delta} - x^{\mathrm{m}}$ (see text) vs $T$ at different $M$ for systems of $N = 16$ XY spins. Here the possible number of quadruplets is $N_q = 1820$. Right : Plot of $x^{\mathrm{M}} - x^{\mathrm{m}}$.

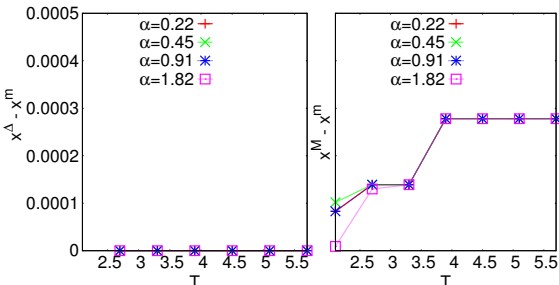

Figure 15: Left/Right: Plot of $x^{\Delta} - x^{\mathrm{m}}/x^{\mathrm{M}} - x^{\mathrm{m}}$ vs $T$ at different $M$ for a system of $N = 32$ XY spin.

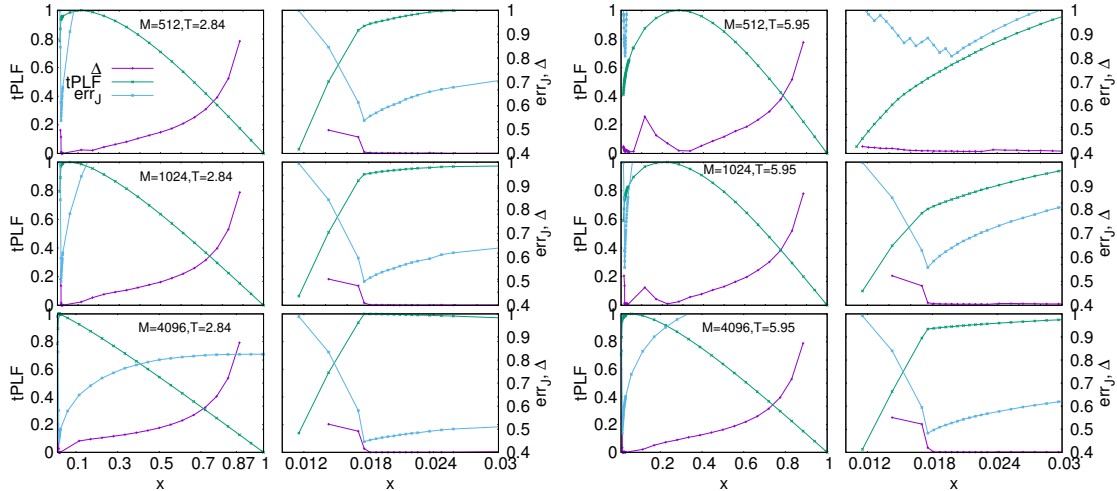

Figure 16: tPLF (green), $\Delta$ (purple) and $\mathrm{err}_J$ (blue) versus $x$. Left: whole range of $x$. Right: zoom in the range of $x$ close to $x^m$, i.e., the minimum of reconstruction error.

In the left hand side we show the results for the entire range of $x \in [0, 1]$. In the right hand side there is a zoom in the $x$ region of interest. Around $T_c$ and/or for high enough $M$, the ideal condition, $x^M = x^m = x^{\Delta}$, is achieved but $x^M$ shifts to higher values for small $M$ and large $T$. On the other hand, $x^{\Delta} = x^m$ until we do not find any discontinuity in the $\Delta$ function.

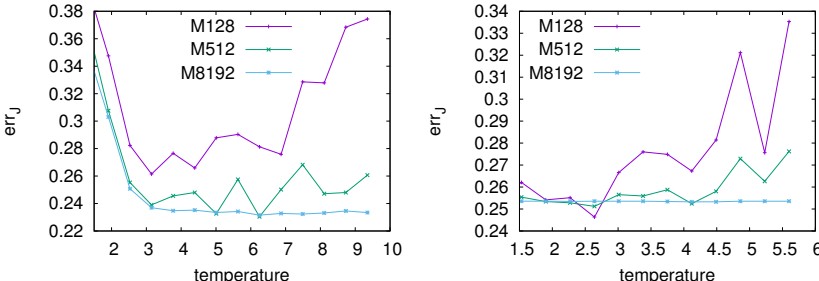

Figure 17: Reconstruction error obtained assuming a $2-XY$ model hypothesis for a system of $N = 16$ XY spins and $N_4 = 32$ quadruplets on an ER-like graph (left) and with $N_4 = 47$ on a ML-like graph (right).

## 5  Pairwise model inference of multi-body systems

What happens when we carry out statistical inference by means of pseudolikelihood maximization if our hypothesis is wrong, i. e., if we maximize with respect to parameters of a wrong model? Are the techniques proposed so far able to identify a wrong theoretical hypothesis? To answer these questions we will analyze some of the previous data, generated by a non-linear - multi-body interacting - system under the hypothesis of pairwise interactions. Hence we will consider an Hamiltonian of the form:

$$\mathcal{H} = \sum_{i<j} J_{ij} \cos\left(\phi_i - \phi_j\right).$$
(38)

On the other hand, we take data from Monte-Carlo simulations of the $4-XY$ model with $N = 16$ spins on a ER graphs. For this system there are initially

$$\binom{N}{4} = \frac{N!}{4!(N-4)!} = 1820$$

quadruplets in total, among which only $N_4 = Nc/2 = 32$ are actually non-zero. The non-zero coupling values are distributed according to a bimodal distribution.

A few words on how we are going to analyze the results of the inference procedure in this case. Indeed, some of the techniques so far exposed rely on the comparison between the inferred network and the original network (TPR, TNR, err$_J$, ...). Some others do not imply any knowledge of the original network (no-match parameter, tPLF($x$) and its maximum in decimation, $\Delta$ function). To use the techniques that rely on the knowledge of the real graph, we convert the system of quadruplets to a system of pairs by converting each quadruplet to 6 pairs. The so generated 6 pairwise bonds will take the same value of the related coupling $J_{ijkl}$. Notice that the same pairwise bond, $J_{ij}$, can pertain to different quadruplets and might thus display different values if the values of the quadruplet couplings are different. As a rule we allow up to 2 pair couplings, out of the 6 related to a single quadruplet, to take a different value. This is enough to build a related pairwise interacting network.

### 5.1  Data analysis

The results are shown for both PLM with $\ell_1$-regularization and with decimation. In Fig. 17 we show the reconstruction error as a function of $T$ for various $M$ for the regularization case. For small $M$ we find a similar pattern as in the previous study and a minimum for $T \sim T_c$ is clearly identified. On the other hand, for high $M$, err$_J$ remains constant above a given $T$.

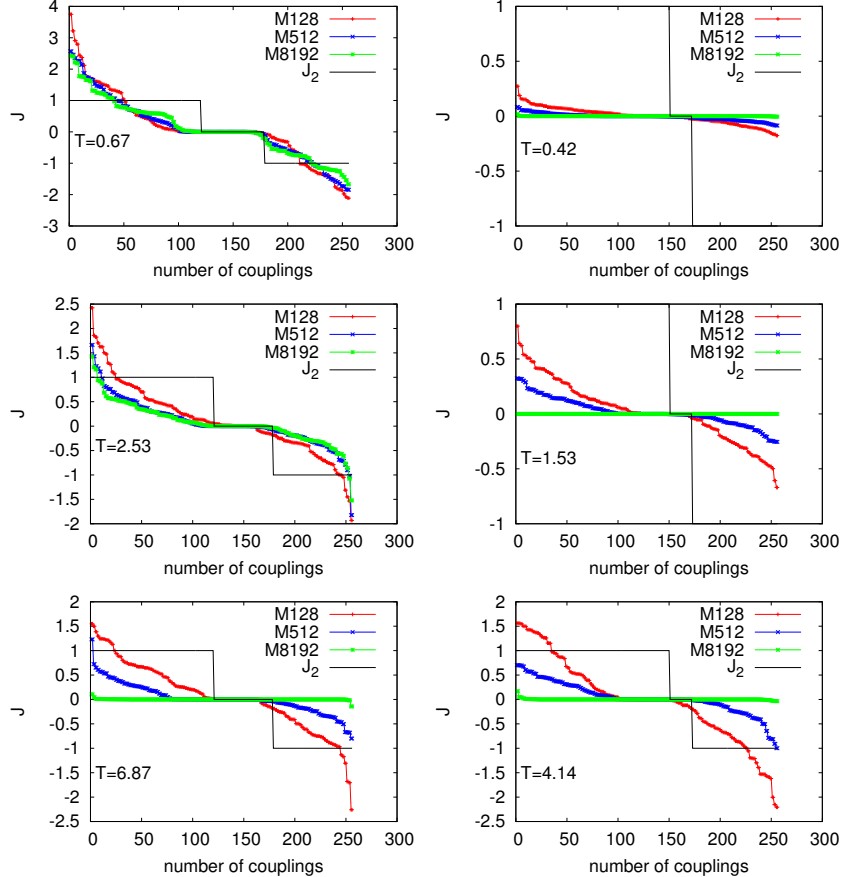

Figure 18: Sorted couplings obtained with the $2 - XY$ model hypothesis through PLM-$\ell_1$. Left: $N = 16$, $N_q = 32$ on a ER-like graph. Right: $N = 16$, $N_4 = 47$ on a ML-like graph.

In Fig. 18, we sort all couplings in descending order. The left plots refer to a system with $N = 16$ XY spins on an ER graph with $N_4 = 32$. In the right, $N = 16$ XY spins interact through $N_q = 47$ quadruplets on a ML graph. In both cases couplings are extracted from a bimodal distribution. The inferred couplings are compared with those of the 2-XY graphs created as described in the previous section (black continuous line in the picture).

Beginning with the top left, we see that at low temperatures the inferred couplings are compatible with a bimodal distribution. Moving to the next panel below, for a higher temperature we see that for larger $M$ the inferred couplings tend to shift more towards zero. This effect is even enhanced in the lowest panel. This trend is observed also in the right column, for the case of ML graph and, once again, it is more evident as the temperature is increased to respect to the critical temperature.

In Fig. 19, we report the results for three values of temperature obtained with PLM with decimation. The tPLF is plotted as a function of the number of non-decimated couplings. We see that at low temperature tPLF curves are overlapping and show a peak at about 100 decimated couplings. Increasing the temperature, the maximum point shifts to lower values. In this case we expect a peak around 20. Increasing the temperature even further, we can see that a clear peak is not even visible and, for high values of $M$, the tPLF does not show an increment remaining around zero: as $M$ increases the t$PLF$ shows that the degrees of freedom used to parametrize the probability of the system configurations are actually irrelevant. In Fig. 20, we show the reconstruction error. In Fig. 21, we show the comparison between the inferred

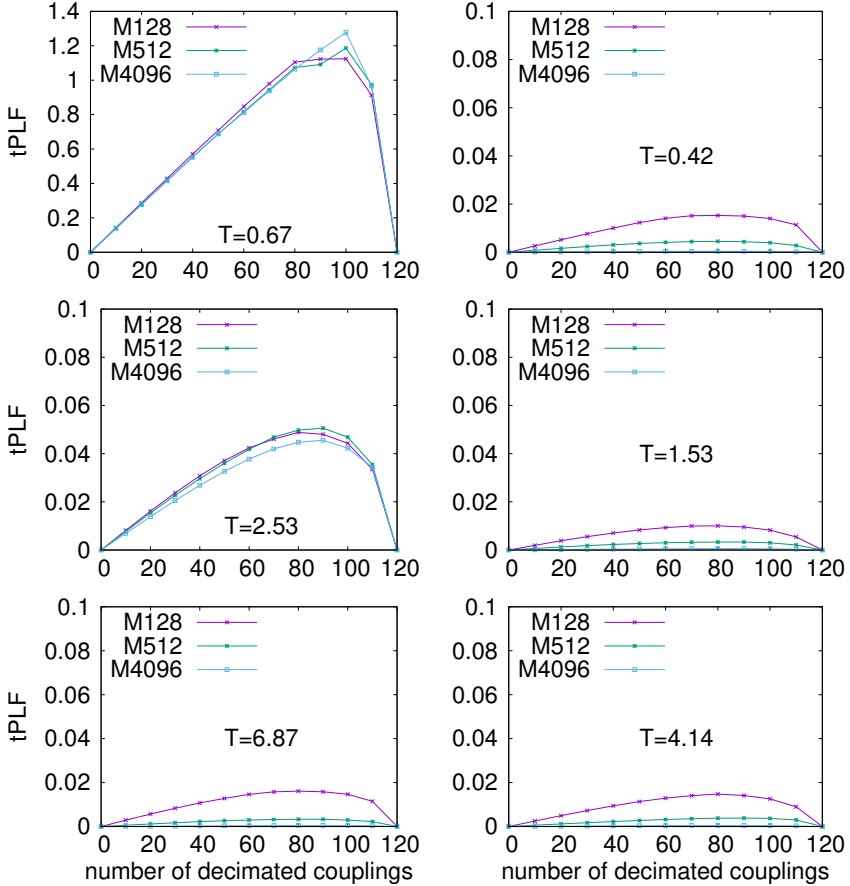

Figure 19: $t$PLF obtained using data of the dynamics of a $4XY$ model with $N = 16$ spins and (left) $N_q = 32$ quadruplets on a ER graph, (right) $N_q = 47$ quadruplets on a ML graph. The PLM algorithm assumes a $2XY$ model: contrary to the results show in previous sections, as $M$ increases the $t$PLF peak smooths down until the function becomes almost independent of the number on decimated couplings.

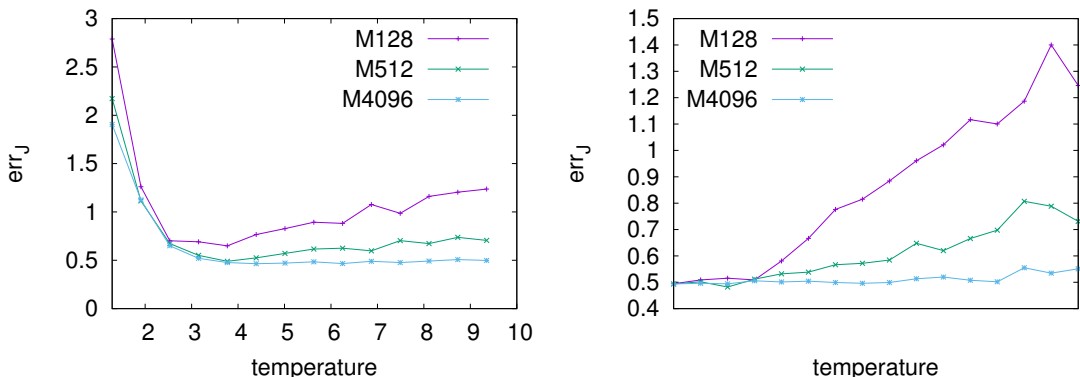

Figure 20: Reconstruction error obtained from $\ell_1$-regularization PLM through PLM with decimation for the systems of Fig. 19.

and the original couplings sorted in ascending order. We find a similar result as with PLM with $\ell_1$-regularization.

From the above observations we have the evidence that if we use a wrong Hamiltonian model as a base for our learning analysis the parameters of the wrong model are simply inferred

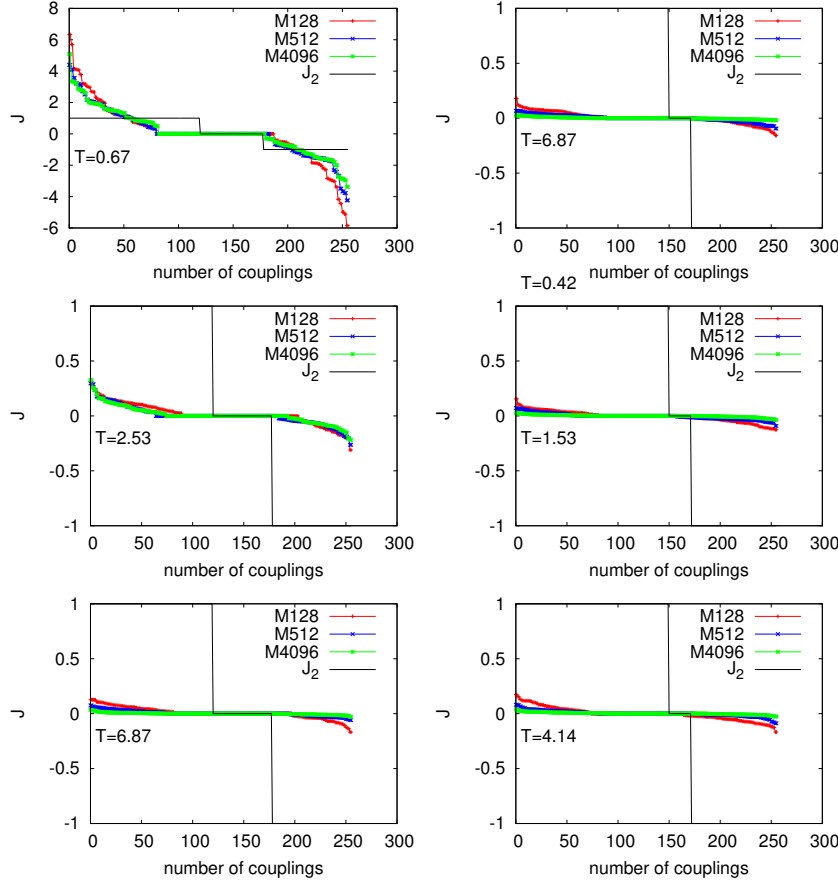

Figure 21: Inferred couplings as obtained from PLM with decimation for the systems of Fig. 19 sorted in ascending order.

to be zero. In other words, the inference procedure is able to find that the wrong model is wrong, that it is not there. The method does not adjust things to adapt to the wrong model yielding some set of non-zero effective pairwise interactions. This discrimination is effective when the external tuning variable $T$ is above the critical point and its quality increases with the number of data series $M$ employed in the procedure.

# 6    Conclusions and future perspectives

In this work, we have presented a deep analysis of the algorithms developed and the results obtained in [50] to solve inverse problems for non-linear continuous spin models. We are motivated by optics: in studying the non-linear interactions among the electromagnetic modes, but the techniques here presented can be applied to a large class of models. In the specific case of non-linear photonic systems, knowledge about the interaction among the modes would give us a proxy to estimate the non-linear optical susceptibility $\chi^{(3)}$ [60]. Further, knowing the frequency associated with each mode we could use statistical inference to probe the existence of phase-locking in random lasers or random media with amplified spontaneous emission (ASE).

In a previous work [15], the Pseudolikelihood algorithm proved to give much better performances with respect to mean field methods for continuous spin models. We have concentrated then on different possible approaches and implementations of the PLM estimator. The results showed that the algorithms are able to reconstruct the network of interactions, with higher

accuracy close to the critical region, as well as the distributions of the couplings.

We also compared the results of the Pseudolikelihood maximization with decimation and with $\ell_1$-regularization. We propose a new criterion to determine the value of the regularizer $\lambda$ for the $\ell_1$ regularization without any a priori knowledge of the system. We analyze the performances of the methods on data generated through Monte-Carlo simulations. We compare the two methods by: i) analyzing the True Positive Rate and the True Negative Rate that provide information on the correctness of the reconstructed interaction graph, ii) studying the reconstruction error that gives information on the relative differences among the inferred and the true couplings, iii) analyzing how good the inferred couplings are in predicting the dynamics of the system, i. e., comparing the true 4-point correlations with those obtained from the dynamics generated with the reconstructed graph. Further, we proposed and verified a new halt criterion for the decimation procedure that allows to achieve better performance for high $T$ and low $M$ when the tilted PLF does not display a clear maximum.

Other interesting questions are still to be addressed. An interesting deep analysis, requiring a work on its own, concerns the robustness against non-equilibrium. Indeed, all the analysis done here is on thermalized data. Future work will analyze the under-sampling regime and the issue of extracting extract useful insights from an under-sampled data set.

## Acknowledgements

We thank Federico Ricci-Tersenghi for fruitful discussions.

**Funding information**    This project has received funding from the European Research Council (ERC) under the European Unions Horizon 2020 research and innovation program, Project LoTGlasSy, Grant Agreement No. 694925 and from the Italian Ministry for Education, University and Research (MIUR) under the PRIN 2015, Project Statistical Mechanics and Complexity, CINECA code 2015K7KK8L_005.

## A  Mode-Locking-like dilution of Erdös-Rényi graphs

Our interest in studying the $XY$ and the complex spherical models resides in optics, with the aim of describing the dynamics of interacting electromagnetic modes in lasers. Previous mean-field studies on fully connected models assume *narrow-band* for the spectrum, [58, 59, 77, 78] that is, all modes practically have the same frequency and, in this way, the frequencies do not play any role in the system behavior. In this section, we will show that non-fully connected graphs can indeed describe the effects of the frequency matching condition intrinsic in mode-locking lasers once the existence of finite-band spectra and gain frequency profiles enter in the description. In particular, if the frequency distribution of the modes is known, we will see how it is possible to derive the remaining interacting quadruplets once the FMC is applied.

In general, we could consider diluted graphs obtained from fully-connected graphs with any kind of dilution. In [22, 23], the authors compared a *homogeneous dilution* (HD), in which the quadruplets are eliminated with some probability that is independent on the mode properties, with a *correlated dilution* (CD), in which the remaining couplings are those among mode quadruplets satisfying the FMC. Firstly, they noticed how to impose the FMC is analogous to introduce a metrics in the problem. Consider the analogy with a random network: the way to construct a graph with a metric is to introduce a distance between different nodes, e.g. $d_{ij} = |i - j|$, and to choose bonds with a probability depending on such a distance. In the case of four-body interactions, one needs a four index function that can be taken as the FMC. In

this way, the mode frequencies play the role of coordinates. Indeed, in Refs. [22, 23] it was quantitatively derived that the closest the modes are in frequency, the highest the probability to be neighbors of the same function node, i.e., to participate to the same quadruplet.

As a starting point, let us consider the case of a Fabry-Pérot cavity laser with a flat gain curve[4]. In this case, the longitudinal resonant frequencies are equispaced with $d\omega = 2\pi/T_R$, being $T_R$ the cavity round trip. We indicate with $N_{\text{quad}}^{in}$ the number of initial quadruplets. We will consider cases in which $N_{\text{quad}}^{in}$ is smaller than the total number of possible quadruplets, $\binom{N}{4}$, but the results here derived can be applied also starting from a fully connected graph. Considering the optical system we have in mind, a first dilution is related to the spatial overlap of the modes: in order to interact the modes have to compete for the same gain medium. For a Fabry-Pérot resonator we expect $N_{\text{quad}}^{in} = \binom{N}{4}$, since all modes fill the entire cavity. For more complicated geometries we would expect $N_{\text{quad}}^{in} < \binom{N}{4}$. In general we may have $\mathcal{O}\left(N_{\text{quad}}^{in}\right) < \mathcal{O}(N^4)$. Starting from $N^{in}$, the quadruplets whose mode frequencies do not satisfy the FMC are erased from the graph. We will term the resulting graphs "Mode-Locking" (ML) graphs.

Knowing the frequency distribution of the modes, we can determine the effect of the FMC on the expected final number of function nodes, $N_{quad}^{out}$. For example, for a multimode Fabry-Pérot cavity we expect a frequency comb spectrum:

$$\omega_n = \omega_0 + (n-1)\delta\omega \qquad \text{with } n = 1,\ldots,N, \tag{39}$$

where $\omega_0$ is some boundary frequency. We consider the realistic case of $\delta\omega/\omega_0 \ll 1$, being in laser $\omega_0$ in the visible light frequency range and $\delta\omega$ in the radio-frequency range. We assume a flat gain curve and one mode for each frequency. For each of the $N_{\text{quad}}^{in}$ we have to verify if the modes belonging to that quadruplet satisfy the FMC. Looking at Eq. (4), i.e.,

$$|\omega_j - \omega_k + \omega_l - \omega_m| \lesssim \gamma,$$

$\gamma$ being the typical linewidth of the mode frequency. We notice that among the 24 possible ordering of the indices $k_1, k_2, k_3, k_4$ in the above expression, can be grouped into 3 independent orderings with 8 equivalent permutations each. Let us consider one ordering among them, which will be indicated with the subindex 1. For example:

$$\text{FMC}_1 : \omega_1 + \omega_3 = \omega_2 + \omega_4 \rightarrow n_1 + n_3 = n_2 + n_4, \tag{40}$$

where we have used Eq. (39) in the last step.

In order to determine the probability for $\text{FMC}_1$ to be satisfied, $P(\text{FMC}_1)$, we can evaluate first the probability distribution of $n_{ij}^+ \equiv n_i + n_j$. Considering the case of uniformly distributed frequencies, i.e., $P(n) = \frac{1}{N}$ for $n \in [1, N]$, we have, with $n^+ \in [2, 2N]$:

$$P^+(n^+) = \begin{cases} \frac{n^+ - 1}{N^2}, & n^+ \in [2, N+1] \\ \frac{2N - (n^+ - 1)}{N^2}, & n^+ \in [N+2, 2N] \end{cases} \tag{41}$$

Then, for the probability that left and right hand side of Eq. (40) be equal we obtain

$$\begin{aligned} P(\text{FMC}_1) &= \sum_{n^+=2}^{2N} P^+(n^+)^2 \\ &= \frac{1 + 2N^2}{3N^3} \sim \frac{2}{3N}, \end{aligned}$$

---

[4]Actually, in Ref. [23], the authors observed that the inclusion of a more complex gain curve affects exclusively the high temperature phase, while the transition and the low temperature phase are stable under perturbation in the gain.

where we are considering the limit $N \gg 1$. The same occurs for $FMC_{2,3}$. Eventually, the number of quadruplets satisfying at least one FMC reads, for large $N$:

$$N_{\text{quad}}^{end} = \frac{2}{N} \left[ 1 + \mathcal{O}\left(\frac{1}{N}\right) \right] N_{\text{quad}}^{in}. \tag{42}$$

Imposing the FMC dilutes a network of $\mathcal{O}(N)$. From this result we learn that in order to obtain a ML diluted graph, with a number of links increasing with $N^\alpha$, we need to start with a denser graph whose number of links scales as $N^{\alpha+1}$.

For instance, in Fig. 22, we show the ML graph obtained starting from an Erdös Rényi graph with $N = 8 \cdot 10^3$ and $N_{quad}^{in} = 76.8 \cdot 10^6 \sim 1.2N^2$. After imposing the FMC we have $N_{quad}^{end} = 2.4N$. The dotted red line represents the connectivity as a function of frequency $\omega$. To show more clearly the behavior in $\omega$, the white full line shows an histogram of the red data. As explained above, before applying the FMC, the frequencies do not play any role, e.g., in fully connected models. On the other hand, in the ML graph, we see that the modes with central frequencies have on average higher values for the connectivity. We have analyzed several cases, with different degrees of dilution in the starting graphs and this frequency dependence of the connectivity has been repeatedly observed. In App. B, we analyze more in details this result deriving the connectivity distribution given the frequency of the node.

In Fig. 22, the black line shows the probability distribution of the connectivity, $P(c)$. In Fig. 23 $P(c)$ is plotted on top of a Poissonian distribution with parameter taken from the mean connectivity of the ML graph. We can see that $P(c)$ coincides with the Poissonian distribution, as explained in the next App. B.

# B Frequency dependence of connectivity in ML graphs

As anticipated in the previous Appendix, in this section we will show that, even after the FMC is imposed, the distribution of the node connectivities, $P_{end}(c(\omega))$, depending now on the frequency $\omega$ of the node, is described by a Poissonian distribution. If we start with an

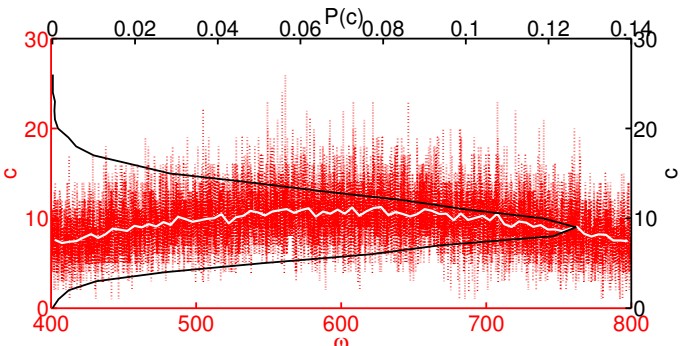

Figure 22: The red dotted line shows the connectivity as a function of frequency of a ML graph. The number of nodes is $N = 8 \cdot 10^3$ and the number of initial quadruplets is $N_{quad}^{in} = 76.8 \cdot 10^6 \simeq 1.2N^2$. The connectivity distribution of this initial graph follows a Poissonian distribution. $N_{quad}^{end} = 19219 \simeq 2N_{quad}^{in}/N$ and the mean connectivity is $\langle c \rangle \simeq 9.6$. We can see that modes with central frequencies have slightly higher connectivity values. The full white line is an histogram of the red data with $N\Delta\omega/\delta b = 80$, being $N\Delta\omega$ the total frequency range and $\delta b$ the bin size. The full black line, $P(c)$, gives the probability distribution of the connectivities; it is plotted with the $c$ values on the $y$-axes to visually enlighten the relation with the red data.

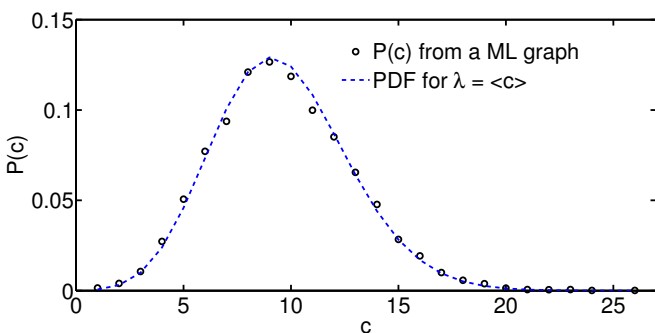

Figure 23: **(b)**: Probability Density Function (PDF) of a Poissonian distribution with parameter $\lambda = \langle c \rangle$, dotted blue line, on top of $P(c)$, empty blue circles: the connectivity distribution of the ML graph is still not very far from the starting Poissonian distribution but with mean connectivity decreased by a factor $2/N$.

Erdös Rényi graph, like in Fig. 22, the distribution of the connectivities of the initial graph will also be Poissonian, with an average $\lambda$ larger of a factor $N$ with respect to the ML graph built from it, cf. Eq. (42). The parameter $\lambda$ of the Poisson distribution is decreased by a factor equal to the probability that, given $\omega$, at least one FMC is satisfied. We indicate the probability with $P_{\text{sat}}(\omega)$; $P_{\text{unsat}}(\omega) = 1 - P_{\text{sat}}(\omega)$ is, on the other hand, the probability that given $\omega$ no FMC is satisfied, i.e., that quadruplet is erased from the graph.

As a first step, let us evaluate the probability that in a ML graph a node does not participate in any quadruplet, i.e., $c(\omega) = 0$. As before, we indicate with $P_{in}(c)$ the probability distribution of the node connectivity before the FMC is applied. We take the ER Poisson case:

$$P_{\text{in}}(c) = \frac{e^{-\lambda}}{c!} \lambda^c, \tag{43}$$

with $\lambda = \langle c \rangle$, independently of $\omega$. For simplicity, we will omit to explicitly write the frequency dependence of $P_{\text{sat}}(\omega)$ and $P_{\text{unsat}}(\omega)$

$$
\begin{aligned}
P_{\text{end}}(0) &= P_{\text{in}}(0) + P_{\text{in}}(1)P_{\text{unsat}} \\
&\quad + P_{\text{in}}(2)P_{\text{unsat}}^2 + \ldots \\
&= \sum_{c=0}^{\infty} P_{\text{in}}(c)P_{\text{unsat}}^c = \sum_{c=0}^{\infty} \frac{e^{-\lambda}}{c!} \lambda^c P_{\text{unsat}}^c \\
&= e^{-\lambda} e^{\lambda P_{\text{unsat}}} = e^{-\lambda P_{\text{sat}}}.
\end{aligned}
\tag{44}
$$

Moving on, the probability that a node in a ML graph will have connectivity one, $P_{\text{end}}(1)$ is:

$$
\begin{aligned}
P_{\text{end}}(1) &= P_{\text{in}}(1)P_{\text{sat}} + 2P_{\text{in}}(2)P_{\text{unsat}}P_{\text{sat}} \\
&\quad + 3P_{\text{in}}(3)P_{\text{unsat}}^2 P_{\text{sat}} + \ldots \\
&= P_{\text{sat}} \sum_{c=0}^{\infty} P_{\text{in}}(c) c \, P_{\text{unsat}}^{c-1} = P_{\text{sat}} \lambda \sum_{c=1}^{\infty} \frac{e^{-\lambda}}{(c-1)!} \lambda^{c-1} P_{\text{unsat}}^{c-1} \\
&= P_{\text{sat}} \lambda e^{-\lambda} e^{-\lambda P_{\text{unsat}}} \\
&= e^{-\lambda P_{\text{sat}}} P_{\text{sat}} \lambda.
\end{aligned}
\tag{45}
$$

Analogously for $P_{\text{end}}(l)$ we obtain:

$$
\begin{aligned}
P_{\text{end}}(l) &= P_{\text{sat}}^l \sum_{c=0}^{\infty} P_{\text{in}}(c) \binom{c}{l} P_{\text{unsat}}^{c-l} \\
&= \frac{(\lambda P_{\text{sat}})^l}{l!} e^{-\lambda P_{\text{sat}}}.
\end{aligned}
\tag{46}
$$

As we can see, we have again a Poissonian distribution with $\lambda \to \lambda P_{sat}$.

The next step is, then, to determine the probability that at least one of the three $\text{FMC}_{1,2,3}$ is satisfied, $P_{\text{sat}}$, i.e.,

$$
P_{\text{sat}} \equiv P\left(|\omega - \omega_{j_1} + \omega_{j_2} - \omega_{j_3}| = 0\right) = P\left(|n - n_{j_1} + n_{j_2} - n_{j_3}| = 0\right),
$$

where we have indicated with $j_{1,2,3}$ three possible modes linked to the node with frequency $\omega = \omega_0 + (n-1)\delta\omega$.

As we did in the previous Appendix, we start by evaluating the probability distribution of $\tilde{P}(\tilde{n}_{j_{1,2,3}} = n_{j_1} - n_{j_2} + n_{j_3})$. Knowing $P(n_{ij}^+ \equiv n_i + n_j)$ from Eq. (48), we have to evaluate:

$$
\tilde{P}(\tilde{n}) = \sum_{n=1}^{N} P(n) P^+(\tilde{n} + n),
\tag{47}
$$

where we used $\tilde{n} = n^+ - n$. We obtain:

$$
\tilde{P}(\tilde{n}) =
\begin{cases}
\frac{1}{N^3} \frac{(\tilde{n}+N)(\tilde{n}+N-1)}{2}, & \tilde{n} \in [2-N, 1] \\
\frac{1}{N^3}\left[(\tilde{n}-1)(N-\tilde{n}) + \frac{N(N+1)}{2}\right], & \tilde{n} \in [2, N] \\
\frac{1}{N^3} \frac{(2N-\tilde{n})(2N-\tilde{n}+1)}{2}, & \tilde{n} \in [N+1, 2N-1]
\end{cases}
\tag{48}
$$

Then, taking into account the three independent ways for $n$ to be equal to $\tilde{n}$ in the argument of $P_{\text{sat}}$, we obtain:

$$
\begin{aligned}
P_{\text{sat}}(\omega = \omega_0 + (n-1)\Delta\omega) &= 3\tilde{P}(n), \quad n \in [1, N] \\
\tilde{P}(n) &= \frac{1}{N^3}\left[(n-1)(N-n) + \frac{N(N+1)}{2}\right].
\end{aligned}
\tag{49}
$$

As we expect, $P_{\text{sat}}(\omega)$ is centered around the central frequency $\omega_c = \omega_0 + \delta\omega(N-1)/2$, but it becomes more and more uniform as $N$ increases.

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
