# Peer review of "Improved Pseudolikelihood Regularization and Decimation methods on Non-linearly Interacting Systems with Continuous Variables"

_SciPost Physics, doi:SciPost Phys. 5, 002 (2018)_

## Round 2 · Referee Report · Pan Zhang (Referee 1) · 2017-11-17

Strengths

Study is systematic and very detailed

Weaknesses

May be a bit lenghy

Report

The authors studied the performance of pseudo-likelihood-based inference of topology of a many-body interacting system. It is clearly shown that decimation method outperforms the method of l-1 regularization, in almost all experiments. I think the paper is well written, and the study is systematic and relevant. However it may be a bit lengthy, I would suggest authors to make the manuscript more compact. Moreover, it would be nice to give some comparisons agains method that is not based on pseudo-likelihood.

Requested changes

In Fig.9, meaning of colors are not specified.

---

## Round 2 · Referee Report · Anonymous (Referee 2) · 2017-11-28

Strengths

This is a detailed work, with careful analysis, on an interesting and timely subject. Methods are rather standard, but new criteria for optimal selection of regularisation parameters are introduced and tested.

Weaknesses

The paper is lengthy, not really well focused, and some important points (applicability of PLM, robustness to perturbations in Hamiltonian) are not discussed enough.

Report

This manuscript focuses on the inference of couplings between laser modes from data. The authors show how pseudo-likelihood methods (PLM) can be used to infer interactions between modes from a set of configurations of the amplitudes of the modes. They propose two variants of PLM and analyse their performance on simulated data.

I have several comments on the paper:

  • it is unclear to me whether the authors’ main objective was to write their paper to report about progress about laser mode physics or about pseudo-likelihood inference. Both are interesting, but the reader may have a hard time to understand from the current formulation what is specific to the details oh Hamiltonian (1) and what is more general here. Please clearly focus on one application, or state clearly what is generic.

  • why the authors refer at several positions in the paper to random laser data, they actually work on simulated data. This raises some issues. For instance, is hypothesis (5) well justified? This is a confusing point as the authors seem to say that (5) is experimentally correct, while they also write right after equation (26) that there are cases where the distinction between zero and small couplings is not easy, which is the case for continuously distributed interactions. Questions: What happens if the data are generated from Hamiltonians where (5) is not exactly satisfied? What happens, more generally, if some random and small Hamiltonian is added to (1) when the data are generated? Section 5 is an attempt to answer partially this question in an extreme case, when the Hamiltonian used to infer parameters is blatantly different from the one used to generate the data. For the special case of pairwise couplings such as in (38), the authors find that all inferred couplings tend to 0 if the number M of data points is sufficiently high. Why is it so? why do not they get some complicated set of effective pairwise interactions, varying with M?

  • general question about the use of PLM close to a transition: there are general necessary conditions for the success of PLM, which worked completely worked out in the Ising case for instance see paper by Ravikumar, Lafferty, Wainwright. In particular, some susceptibility matrix must have a norm smaller than unity to avoid amplifying errors during the iterative maximisation of the pseudo-likelihood. Are these conditions satisfied here, even above the transition temperature where the author operate (see Figure 4 for instance)?

  • section 4 is very hard to read as it is lengthy, and report many results of the inference procedure applied to a variety of cases. Could the author rewrite it in a more synthetic way, extracting only meaningful results and messages for the readers?

Minor comments:

Beginning of Section 3: I do not understand how the omega_k were generated. Are they drawn from some distribution? If so, how is the latter chosen? I could find only a brief sentence about this point in Appendix A, right before equation (40). This is an important point regarding FMC.

Sentence right after equation (24): “We note that in the mean-field case, one crucial minimal criterion for the inverse problem to be tractable is M to be equal to N since the correlation matrix needs to be invertible. In the present method this lower bound is not strictly requested.” This is not correct, mean-field inference is ok as soon as M>=N.

Section 4.1: please give explicit formula for the no-match parameter. It is hard to understand how it is precisely defined.

There are many typos and spelling mistakes. Please correct.

Requested changes

See report

---

## Round 3 · Referee Report · Anonymous (Referee 3) · 2018-3-14

Report

I think the revised version is now publishable.

---

## Round 3 · Referee Report · Anonymous (Referee 2) · 2018-3-30

Report

I think that the manuscript has been improved with respect to the previous version. Section 4, in particular, is more accessible, and some points (see reports) have been clarified.

Prior to publication, I would appreciate if the authors could again consider the comment I made in my initial report, that is, point 4 in their rebuttal letter. It is hard for me to understand why inference with a wrong model leads to vanishing couplings, and not to something complicated and essentially inconsistent, but that fits the data! I find the answer rather not convincing on this aspect. Let me be clear: I am not saying that the authors are wrong and their study is incorrect, I am simply saying that I do not understand their findings and that they do not present compelling arguments allowing me to get what is going on. If I take a bunch of data generated by model 1 and fit with model 2, I will get a set of meaningless parameters for model 2, fitting as best as possible the data. Why should these parameter vanish? It will depend on the expressive power of model 2 with respect to model 1, won't it?
I think the paper would benefit if some comments on this point could be inserted prior to publication.

---

## Round 3 · Author Response

Reply to referee 1, Pan Zhang

We thank the referee for his review of our manuscript and his comments. We recognize that the previous version was actually lengthy and we made an effort to shorten the revised version.

Pan asks us to give comparisons against other methods, not based of the PLM. In previous papers, though (P. Tyagi et al. JSTAT 2015 and PRB 2016, Refs. [15,31] in the revised version), for pairwise interacting systems, we studied other estimators based on mean field approximations and compared their performances to those of the pseudolikelihood maximization estimators. With the PLM we obtained better performances even in the low sampling regime. As rightly pointed out by the referee in his comments, in this work we did not repeat the comparison to other standard methods but we concentrate on techniques based on the pseudolikelihood maximization. We, however, notice that, in the cases under study, in which strong non-linearities and multi-body interactions occur among variables, techniques other than PLM-based ones would be even more computationally demanding with respect to the cases already analyzed in Refs. [15,31]. To clarify this point to the readers we explicitly write, in the conclusions: “In a previous\cite{Tyagi16} work, the Pseudolikelihood algorithm proved to give much better performances with respect to mean field methods for continuous spin models. We have concentrated then on different possible approaches and implementations of the PLM estimator. The results showed that the algorithms are able to reconstruct the network of interactions, with higher accuracy close to the critical region, as well as the distributions of the couplings.”

We added in Fig. 9 caption the specifications of blue and red colours to correlations computed on differently inferred system networks.

Reply to referee 2, We thank the referee for his/her careful analysis of our work and we herewith reply to his/her comments.

1) The referee comments “it is unclear to me whether the authors’ main objective was to write their paper to report about progress about laser mode physics or about pseudo-likelihood inference. Both are interesting, but the reader may have a hard time to understand from the current formulation what is specific to the details oh Hamiltonian (1) and what is more general here. Please clearly focus on one application, or state clearly what is generic.”

This is a very important issue, and we thank the referee for pointing out at it, since our original motivation (to understand random laser behaviour and/or nonlinear effects contribution to light propagation in random media) may have generated confusions about the machine learning methods that we analyze. The paper is about progress on pseudo likelihood inference. In particular, on systems whose interactions are not pairwise, but multi-body. To clarify this, in the introduction, on the first paragraph of page 3 of the revised version we now write: “Even though our original motivation arises in the framework of nonlinear optics and laser physics [14,20-24], in this work we will concentrate on state-of-the-art techniques to solve inverse problems. Indeed, the techniques here presented can be applied to a large class of models with multi-body interactions.”

Results are generic, but we test the methods on a set of models (diverse variables, qualitatively diverse network connectivities). These are introduced in an apart Section 2 “Test Model”, where we spend a few lines reporting how the models stem out from the study of nonlinearly interacting waves. However, we keep this part, interesting to both nonlinear optics and statistical mechanics (sub)communities, conceptually separated from the rest. To clarify this point we begin Sec. 2 with the head sentence: “In this section the models and the physical systems of interest are introduced. The interested reader can find more details in [14]. This paper is organized in such a way to let the reader interested only in inference techniques applied to nonlinear systems to skip this section.”

2) The referee asks “why the authors refer at several positions in the paper to random laser data, they actually work on simulated data. This raises some issues. For instance, is hypothesis (5) well justified? This is a confusing point as the authors seem to say that (5) is experimentally correct, while they also write right after equation (26) that there are cases where the distinction between zero and small couplings is not easy, which is the case for continuously distributed interactions.”

Actually, nowhere we claim that Eq. (5) is experimentally satisfied for all possible materials and applications or that it is the only possible distribution of couplings. We have added a small paragraph to explain more in details why we choose Eq. 5, as well as a Gaussian distribution, as probability distributions for the coupling costants: “ Because of the partial knowledge of modes localization and the very poor knowledge of the nonlinear response so far in experiments, the random values for the $J$s can be taken from any physically reasonable arbitrary probability distribution. We will then take couplings of a multibody interacting network, with number of coupling nodes $N_q$ scaling as $N_q \sim N^z$ with the number of variables $N$, as generated through a bimodal distribution, i.e., \begin{equation} P(J)=1/2[\delta(J-\hat J)+\delta(J+\hat J)], \end{equation} with $\hat J=1/N^{(z-1)/2}$, or through a Normal Gaussian distribution of mean square displacement $\sigma \sim \hat{J}$. Indeed, we can analyze the performance of the inference techniques for both discrete and continuously distributed couplings. “

Bimodal stands for a test on discrete couplings, Gaussian for a test on continuous couplings. We simulate both systems, and analyse both kind of data with the same techniques. Our aim is to show the performances of the methods treated in both cases, for the most general application.

3) The referee asks “Questions: What happens if the data are generated from Hamiltonians where (5) is not exactly satisfied?”

The PLM methods we expose reproduce the coupling values however generated. That is, actually, the aim: to provide a machine learning procedure to infer unknown couplings. Therefore we test the methods using Eq. (5) for the J values - thus Bimodal, using Gaussian distributions with zero average, using constant values of the couplings. On top of the values also the networks are random and we generated networks with different connectivities, scaling like N, N^2, or N^3. Also the local topologies we have adopted for the networks are of two different kinds: one is uniformly random (poissonian-like and , thus, termed Redos-Renyi), one is based on a mode-locking rule, associating quenched “frequencies” to the variables. The difference smoothens in the thermodynamic limit but it is rather important at finite N, as reported in Appendices A and B.

4) “What happens, more generally, if some random and small Hamiltonian is added to (1) when the data are generated? Section 5 is an attempt to answer partially this question in an extreme case, when the Hamiltonian used to infer parameters is blatantly different from the one used to generate the data. For the special case of pairwise couplings such as in (38), the authors find that all inferred couplings tend to 0 if the number M of data points is sufficiently high. Why is it so? why do not they get some complicated set of effective pairwise interactions, varying with M?”

What happens depends on the random Hamiltonian, that is unknown in the real world. If we have a 4-body interacting system inferred via a 4-body interacting model any 4-body perturbation to it is automatically taken into account, as it is clear from the answer to question 3. If the original system has, instead, say, linear perturbations, or more generically is a “2+4” interacting model, than we need a 2+4 Hamiltonian (see, e.g., Refs. [20,21]) to “catch” all couplings. We did not go through this analysis in the present paper because it would be a lengthy analysis and requires expensive simulations to generate data, but we do not need to do that to display the power of PLM-based methods because the core of the result is already in the analysis of the blatantly wrong Hamiltonian of Sec. 5. There, we show that if we use a wrong Hamiltonian model, and we have enough data to analyze, the parameters of the wrong model are simply inferred to be zero. In other words, the inference procedure is able to find that the wrong model is wrong, it is not there. The method does not adjust things to adapt to the wrong model yielding “some complicated set of effective pairwise interactions”. Having proper experimental data (that we do not have), therefore, and using a generic model, say a combined 2+3+4 model combining, respectively, linearly interacting waves and non linearly interacting waves with both possible non-centrosymmetric potential (3-body) and centrosymmetric potential (4-body), would rule out inexistent contributions (setting them to zero) and only infer the values of existing contributions. With no priori knowledge. And the quality of this discrimination increases with M and with the range of external tuning variables such as the temperature. Indeed, the risky statistical artefact of inferring “some complicated set of effective pairwise interactions” appears to be washed out by the PLM if M is large enough.

5) Furthermore, the referee asks a “general question about the use of PLM close to a transition: there are general necessary conditions for the success of PLM, which worked completely worked out in the Ising case for instance see paper by Ravikumar, Lafferty, Wainwright. In particular, some susceptibility matrix must have a norm smaller than unity to avoid amplifying errors during the iterative maximisation of the pseudo-likelihood. Are these conditions satisfied here, even above the transition temperature where the author operate (see Figure 4 for instance)?”

Concerning this point we added a small paragraph in Section "Improved Pseudo Likelihood Maximization with $l_1$ regularization: hypothesis testing" where we explained the requirements the Fisher Information Matrix needs to satisfy in order for PLM with l1-reg to have a unique solution and to correctly reconstruct the neighbors of a node: "Note that, in order for the procedure to be consistent, the eigenvalues of the Fisher Information Matrix needs to be bounded from below. This condition together with the requirement that the entries of $\mathcal{I}^i_{a b}$ related to non-neighbors of $i$ cannot exercise an overly strong effect on the subset related to the neighbors of $i$ assures that the PLM with $l_1$ regularization has a unique solution and correctly reconstruct the neighbors of $i$ if enough number of samples are provided\cite{Ravikumar10}."

The first requirement is always checked since we evaluate the eigenvalues and, in order for the procedure to be consistent, they have to be bounded. The second requirement is checked a posteriori.

6) “section 4 is very hard to read as it is lengthy, and report many results of the inference procedure applied to a variety of cases. Could the author rewrite it in a more synthetic way, extracting only meaningful results and messages for the readers?”

We agree with the referee. We tried our best to shorten Sec. 4 and make easier to read in the revised version.

---

## Round 3 · List of Changes

In general the text has been sensitively shortened, as required by both referees.

Further changes are the following.

Introduction:
On page 3 we add the paragraph:
“Even though our original motivation arises in the framework of nonlinear optics and laser physics [14,20-24], in this work we will concentrate on state-of-the-art techniques to solve inverse problems. Indeed, the techniques here presented can be applied to a large class of models with multi-body interactions.”

Sec. 2:
after Eq. (5) we modify the text as:
“ Because of the partial knowledge of modes localization and
the very poor knowledge of the nonlinear response so far in experiments, the random
values for the $J$s can be taken from any physically reasonable arbitrary probability distribution.
We will then take couplings of a multibody interacting network, with number of coupling nodes $N_q$ scaling as $N_q \sim N^z$ with the number of variables $N$, as generated through a bimodal distribution, i.e.,
\begin{equation}
P(J)=1/2[\delta(J-\hat J)+\delta(J+\hat J)],
\end{equation}
with $\hat J=1/N^{(z-1)/2}$,
or through a Normal Gaussian distribution of mean square displacement
$\sigma \sim \hat{J}$.
Indeed, we can analyze the performance of the inference techniques for both discrete and continuously distributed couplings. “

Sec. 3.2:
The paragraph has been added:
"Note that, in order for the procedure to be consistent, the eigenvalues of the Fisher Information Matrix needs to be bounded from below. This condition together with the requirement that the entries of $\mathcal{I}^i_{a b}$ related to non-neighbors of $i$ cannot exercise an overly strong effect on the subset related to the neighbors of $i$ assures that the PLM with $l_1$ regularization has a unique solution and correctly reconstruct the neighbors of $i$ if enough number of samples are provided\cite{Ravikumar10}."

Fig. 9 caption: color references added

Conclusions:
paragraph added: “In a previous\cite{Tyagi16} work, the Pseudolikelihood algorithm proved to give much better performances with respect to mean field methods for continuous spin models. We have concentrated then on different possible approaches and implementations of the PLM estimator. The results showed that the algorithms are able to reconstruct the network of interactions, with higher accuracy close to the critical region, as well as the distributions of the couplings.”

---

## Round 4 · Referee Report · Anonymous (Referee 5) · 2018-5-30

Report

I am grateful to the authors for their explanations, and I think the manuscript can now be published as it stands.

---

## Round 4 · Author Response

We thank referee 2 for her/his observations and we give a detailed answer in the following.

The Referee comments:

"Prior to publication, I would appreciate if the authors could again consider the comment I made in my initial report, that is, point 4 in their rebuttal letter. It is hard for me to understand why inference with a wrong model leads to vanishing couplings, and not to something complicated and essentially inconsistent, but that fits the data! I find the answer rather not convincing on this aspect. Let me be clear: I am not saying that the authors are wrong and their study is incorrect, I am simply saying that I do not understand their findings and that they do not present compelling arguments allowing me to get what is going on. If I take a bunch of data generated by model 1 and fit with model 2, I will get a set of meaningless parameters for model 2, fitting as best as possible the data. Why should these parameter vanish? It will depend on the expressive power of model 2 with respect to model 1, won't it?

Authors reply:

The referee gives for granted that "if I take a bunch of data generated by model 1 and fit with model 2, I will get a set of meaningless parameters for model 2, fitting as best as possible the data". Even though this can be a possible outcome, it is not the only outcome of an inference procedure. Another possibility is to get meaningful parameters for model 2, i. e., parameters tending to zero. In particular, in our analysis above the critical point of our model, the outcome provides small pairwise couplings, decreasing as $M$ is increased.

We try to clarify the results obtained in a more formal way. Let us first stress that, when the temperature is above the critical temperature, in a $4$-body system all two point correlations functions are zero even if the two variables belong to the same quadruplet. On the contrary, four point correlations of four variables in the same interacting quadruplet are non-zero. To observe the consequence of the correlation between $i$ and $j$ being zero on the value of the coupling $J_{ij}$ between them, let us compute the two point correlation explicitly in the wrong pairwise hypothesis. We assume a Hamiltonian of the form Eq. (37):

$$H = -\sum_{i,j \in E_{ij} } J_{ij} \cos{\left(\phi_i - \phi_j\right)}$$
where $E_{ij}$ indicates the edges actually present in the graph (an edge is there when $J_{ij}\neq 0$). In this case the Pseudolikelihood $P_i(\phi_i | \gv{\phi}_{\backslash i})$ is:
$$ P_i(\phi_i | \gv{\phi}{\backslash i}) = \frac{1}{Z_i} \exp{\bigl[H_i^x \left(\gv{\phi}\right) \cos(\phi_i) + H_i^y \left(\gv{\phi}_{\backslash i}\right) \sin(\phi_i)\bigr] }$$
where we use the shortening
$$ H^x_i(\gv{\phi}{\backslash i}) := \sum J_{ij}\cos(\phi_j)$$
$$ H^y_i(\gv{\phi}{\backslash i}) := \sum J_{ij}\sin(\phi_j)$$
and
$$ Z_i = 2 \pi I_0( H_i)$$
$$ H_i \equiv \sqrt{(H^x_i)^2 + (H^y_i)^2}$$
where $I_0$ is the modified Bessel function defined in Sec. 3.1.1 and $\partial i$ denotes the neighbors if $i$. {We have rescaled the $J$s to include $\beta$.}

Let us consider the case in which we are given $M$ independent observations ${\gv{\phi}^{(\alpha)}}_{\alpha=1}^M$. The log-pseudolikelihood is defined as: \begin{eqnarray} \label{eq:pl} L_i & := &\frac{1}{M}\sum_{\alpha=1}^M\log{ P_i(\phi^{(\alpha)}i | \gv{\phi}^{(\alpha)})}\ & = & \frac{1}{M}\sum_{\alpha=1}^M \left{H_i^x \left(\gv{\phi^\alpha}{\backslash i}\right) \cos(\phi^\alpha_i) + H_i^y \left(\gv{\phi^\alpha}\right) \sin(\phi^\alpha_i) - \ln 2 \pi I_0(H_i^\alpha)\right} \nonumber \end{eqnarray}

Within the decimation procedure we minimize ${-}\sum_i L_i$. Evaluating the first derivative with respect to $J_{l m}$ and setting the result to zero, we obtain the equation \begin{eqnarray} \label{eq:corr_jj} \sum_{\alpha=1}^M \cos{\left(\phi^\alpha_l - \phi^\alpha_m\right)} = \frac{1}{2}\sum_{\alpha=1}^{M} && !!! \Biggl[\frac{I_1(H_l^\alpha)}{I_0(H_l^\alpha)}\frac{\cos \phi^\alpha_m H^x_l+\sin \phi^\alpha_m H^y_l}{H^\alpha_l}
\ \nonumber && + \frac{I_1(H_m^\alpha)}{I_0(H_m^\alpha)}\frac{\cos \phi^\alpha_l H^x_m+\sin \phi^\alpha_l H^y_m}{H^\alpha_m}\Biggr] \end{eqnarray}

Recalling that $I_0(x)\to 1$ and $I_1(x) \sim x$ for small $x$, we find that for small $J$'s

\begin{equation} \sum_{\alpha=1}^M \cos{\left(\phi^\alpha_l - \phi^\alpha_m\right)} \simeq \sum_{\alpha}^{M}\left[\cos \phi^\alpha_m H^x_l+\sin \phi^\alpha_m H^y_l + \cos \phi^\alpha_l H^x_m+\sin \phi^\alpha_l H^y_m \right] \end{equation} In the paramagnetic phase of the $4$-body Hamiltonian system, that is above the critical temperature, the correlation between the nodes $l$ and $m$ in the left hand side turns out to be zero. Therefore, if $M$ is high enough, we expect the right hand side of Eq. \eqref{eq:corr_jj} to be zero, as well: all the $J$'s will move to lower and lower values as $M$ increases and eventually tend to zero.

Fig. 18 reflects this result for the ML-graph (right) for which we are always above $T_c \sim 0.5$.

On the contrary, for the ER-graph (left panel of Fig. 18) - for which $T_c \sim 2.3$ - we see that the algorithm tries to accomodate to some effective non-zero $J$'s that can describe $\overline{\cos{\left(\phi_l-\phi_m\right)}}$ at $T\lesssim T_c$. The same behavior is shown in Fig. 19.

---

## Round 4 · List of Changes

Section 5. Last paragraph added.

Appendices: minor changes

Ref. [78] replaced.

---

## Editorial Decision

published